# Modelling sequences and temporal networks with dynamic community structures

Tiago P. Peixoto[1,2] & Martin Rosvall[3]

In evolving complex systems such as air traffic and social organisations, collective effects emerge from their many components' dynamic interactions. While the dynamic interactions can be represented by temporal networks with nodes and links that change over time, they remain highly complex. It is therefore often necessary to use methods that extract the temporal networks' large-scale dynamic community structure. However, such methods are subject to overfitting or suffer from effects of arbitrary, a priori-imposed timescales, which should instead be extracted from data. Here we simultaneously address both problems and develop a principled data-driven method that determines relevant timescales and identifies patterns of dynamics that take place on networks, as well as shape the networks themselves. We base our method on an arbitrary-order Markov chain model with community structure, and develop a nonparametric Bayesian inference framework that identifies the simplest such model that can explain temporal interaction data.

[1] Department of Mathematical Sciences and Centre for Networks and Collective Behaviour, University of Bath, Claverton Down, Bath BA2 7AY, UK. [2] ISI Foundation, Via Alassio 11/c, 10126 Torino, Italy. [3] Integrated Science Lab, Department of Physics, Umeå University, SE-901 87 Umeå, Sweden. Correspondence and requests for materials should be addressed to T.P.P. (email: t.peixoto@bath.ac.uk)

To reveal the mechanisms of complex systems, researchers identify large-scale patterns in their networks of interactions with community-detection methods[1]. Traditionally, these methods describe only static network structures without taking into account the dynamics that take place on the networks, such as people travelling by air, or the dynamics of the networks themselves, such as new routes in air traffic networks. While the dynamics on and of networks contain crucial information about the systems they represent, only recently have researchers showed how to incorporate higher-order Markov chains to describe dynamics on networks[2–5] and higher-order temporal structures to describe dynamics of networks[6–18]. However, both avenues of research have encountered central limitations: first, methods that use higher-order memory to describe dynamics on networks rely on extrinsic methods to detect the appropriate memory order[2,19]. Second, methods that attempt to describe dynamics of networks adapt static descriptions by aggregating time windows into discrete layers[13–15,17,20–22], and ignore dynamics within the time windows. Thus, both methods for dynamics on and of networks require or impose ad hoc timescales that can obscure essential dynamic community structure.

Furthermore, when trying to determine the timescales solely from data, the curse of dimensionality strikes: the large number of degrees of freedom makes the higher-order descriptions prone to overfitting, when random fluctuations in high-dimensional data are mistaken for actual structure[23]. Without a principled method with effective model selection to counteract this increasing complexity, it becomes difficult to separate meaningful dynamic community structure from artefacts.

To overcome these model selection and arbitrary timescale problems, we present a general and principled data-driven method by simultaneously tackling dynamics on and of networks (Fig. 1). In contrast to approaches that incorporate temporal layers in methods for static network descriptions, we build our approach on describing the actual dynamics. We first formulate a generative model of discrete temporal processes based on arbitrary-order Markov chains with community structure[24–27]. Since our model generates event sequences, it does not aggregate data in time windows[13–15,17], and, other than the Markov model assumption, needs no a priori imposed timescales. This model can be used to describe dynamics taking place on network systems that take into account higher-order memory effects[2,3] of arbitrary order. We then use the model to describe temporal networks, where the event sequence represents the occurrence of edges in the network[10].

In both cases, we employ a nonparametric Bayesian inference framework that allows us to select, according to the statistical evidence available, the most parsimonious model among all its variations. Hence we can, for example, identify the most appropriate Markov order and the number of communities without overfitting. In particular, if the dynamics on or of a network are random, our method will not identify any spurious patterns from noise but conclude that the data lack structure. As we also show, the model can be used to predict future network dynamics and evolution from past observations. Moreover, we provide publicly available and scalable code with log-linear complexity in the number of nodes independent of the number of groups.

## Results

**Inference of markov chains.** Here we consider general time-series composed of a sequence of discrete observations $\{x_t\}$, where $x_t$ is a single token from an alphabet of size $N$ observed at discrete time $t$, and $\boldsymbol{x}_{t-1} = (x_{t-1}, \ldots, x_{t-n})$ is the memory of the previous $n$ tokens at time $t$ (Fig. 1). An $n$th-order Markov chain with transition probabilities $p(x_t|\boldsymbol{x}_{t-1})$ generates such a sequence with probability

$$P(\{x_t\}|p) = \prod_i p(x_t|\boldsymbol{x}_{t-1}) = \prod_{x,\boldsymbol{x}} p(x|\boldsymbol{x})^{a_{x,\boldsymbol{x}}}, \quad (1)$$

where $a_{x,\boldsymbol{x}}$ is the number of transitions $\boldsymbol{x} \to x$ in $\{x_t\}$. Given a specific sequence $\{x_t\}$, we want to infer the transitions probabilities $p(x|\boldsymbol{x})$. The simplest approach is to compute the maximum-likelihood estimate, that is

$$\widehat{p}(x|\boldsymbol{x}) = \underset{p(x|\boldsymbol{x})}{argmax}\, P(\{x_t\}|p) = \frac{a_{x,\boldsymbol{x}}}{a_{\boldsymbol{x}}}, \quad (2)$$

where $a_{\boldsymbol{x}} = \sum_x a_{x,\boldsymbol{x}}$, which amounts simply to the frequency of observed transitions. Putting this back into the likelihood of eq. (1), we have

$$\ln P(\{x_t\}|\widehat{p}) = \sum_{x,\boldsymbol{x}} a_{x,\boldsymbol{x}} \ln \frac{a_{x,\boldsymbol{x}}}{a_{\boldsymbol{x}}}. \quad (3)$$

This can be expressed through the total number of observed transitions $E = \sum_{x,\boldsymbol{x}} a_{x,\boldsymbol{x}}$ and the conditional entropy $H(X|\mathbf{X}) = -\sum_{\boldsymbol{x}} \hat{p}(\boldsymbol{x}) \sum_x \hat{p}(x|\boldsymbol{x}) \ln \hat{p}(x|\boldsymbol{x})$ as $\ln P(\{x_t\}|\hat{p}) = -EH(X|\mathbf{X})$. Hence, the maximisation of the likelihood in eq. (1) yields the transition probabilities that most compress the sequence. There is, however, an important caveat with this approach. It cannot be used when we are interested in determining the most appropriate Markov order $n$ of the model, because the maximum likelihood in eq. (3) increases with $n$. In general, increasing number of memories at fixed number of transitions leads to decreased conditional entropy. Hence, for some large enough value of $n$ there will be only one observed transition conditioned on every memory, yielding a zero conditional entropy and a maximum likelihood of 1. This would be an extreme case of overfitting, where by increasing the number of degrees of freedom of the model it is impossible to distinguish actual structure from stochastic

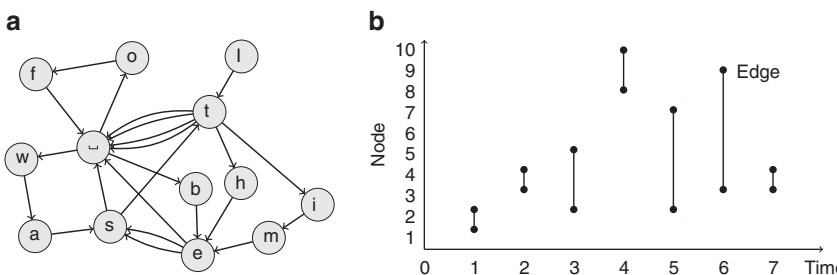

**Fig. 1** Unified modelling of dynamics on and of networks. Our modelling framework simultaneously describes: **a** Arbitrary dynamics taking place on networks, represented as a sequence of arbitrary tokens that are associated with nodes, in this example $\{x_t\}$ = 'It-was the best of times'. **b** Dynamics of networks themselves, where the tokens are node-pair edges that appear in sequence, in this example $\{x_t\}$ = {(1, 2), (4, 3), (5, 2), (10, 8), (7, 2), (9, 3), (3, 4)}

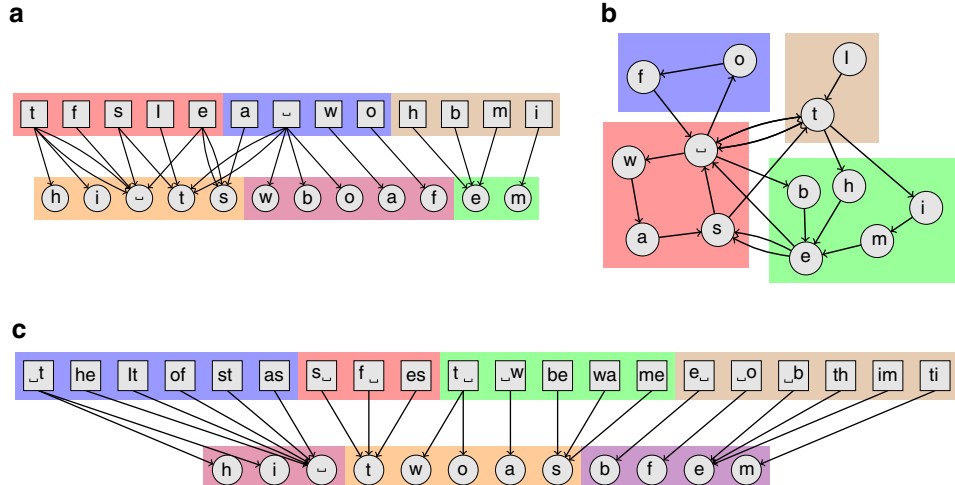

**Fig. 2** Schematic representation of the Markov model with communities. The token sequence $\{x_t\}$ = 'It was the best of times' represented with nodes for memories (*top row*) and tokens (*bottom row*), and with directed edges for transitions in different variations of the model. **a** A partition of the tokens and memories for an $n = 1$ model. **b** A unified formulation of an $n = 1$ model, where the tokens and memories have the same partition, and hence can be represented as a single set of nodes. **c** A partition of the tokens and memories for an $n = 2$ model

fluctuations. Also, this approach does not yield true compression of the data, since it does not describe the increasing model complexity for larger values of $n$, and thus is crucially incomplete. To address this problem, we use a Bayesian formulation, and compute instead the complete evidence

$$P(\{x_t\}) = \int P(\{x_t\}|p)P(p)\,\mathrm{d}p, \qquad (4)$$

which is the sum of all possible models weighted according to prior probabilities $P(p)$ that encode our a priori assumptions. This approach gives the correct model order for data sampled from Markov chains as long as there are enough statistics that balances the structure present in the data with its statistical weight, as well as meaningful values when this is not the case[28].

Although this Bayesian approach satisfactorily addresses the overfitting problem, it misses opportunities of detecting large-scale structures in data. As we show below, it is possible to extend this model in such a way as to make a direct connection to the problem of finding communities in networks, yielding a stronger explanatory power when modelling sequences, and serving as a basis for a model where the sequence itself represents a temporal network.

**Markov chains with communities**. Instead of directly inferring the transition probabilities of eq. (1), we propose an alternative formulation: We assume that both memories and tokens are distributed in disjoint groups (Fig. 2). That is, $b_x \in [1, 2, ..., B_N]$ and $b_x \in [B_N + 1, B_N + 2, ..., B_N + B_M]$ are the group memberships of the tokens and memories uniquely assigned in $B_N$ and $B_M$ groups, respectively, such that the transition probabilities can be parametrised as

$$p(x|\boldsymbol{x}) = \theta_x \lambda_{b_x b_x}. \qquad (5)$$

Here $\theta_x$ is the relative probability at which token $x$ is selected among those that belong to the same group, and $\lambda_{rs}$ is the overall transition probability from memory group $s = b_x$ to token group $r = b_x$. The parameter $\theta_x$ plays an analogous role to degree-correction in the SBM[29], and is together with the Bayesian description the main difference from the sparse Markov chains developed in refs. [26, 27]. In the case $n = 1$, for example, each token appears twice in the model, both as token and memory. An

alternative and often useful approach for $n = 1$ is to consider a single unified partition for both tokens and memories, as shown in Fig. 2b and described in detail in the Methods section 'The unified first-order model'. In any case, the maximum likelihood parameter estimates are

$$\hat{\lambda}_{rs} = \frac{e_{rs}}{e_s}, \quad \hat{\theta}_x = \frac{k_x}{e_{b_x}}, \qquad (6)$$

where $e_{rs}$ is the number of observed transitions from group $s$ to $r$, $e_s = \sum_t e_{ts}$ is the total outgoing transitions from group $s$ if $s$ is a memory group, or the total incoming transition if it is a token group. The labels $r$ and $s$ are used indistinguishably to denote memory and token groups, since it is only their numerical value that determines their kind. Finally, $k_x$ is the total number of occurrences of token $x$. Putting this back in the likelihood, we have

$$\ln \hat{P}(\{x_t\}|b, \hat{\lambda}, \hat{\theta}) = \sum_{r < s} e_{rs} \ln \frac{e_{rs}}{e_r e_s} + \sum_X k_x \ln k_x. \qquad (7)$$

This is almost the same as the maximum likelihood of the degree-corrected stochastic block model (DCSBM)[29], where $a_{x,x}$ plays the role of the adjacency matrix of a bipartite multigraph connecting tokens and memories. The only differences are constant terms that do not alter the position of the maximum with respect to the node partition. This implies that for undirected networks without higher-order memory, there is no difference between inferring the structure directly from its topology or from dynamical processes taking place on it, as we show in detail in the Methods section Equivalence between structure and dynamics.

As before, this maximum likelihood approach cannot be used if we do not know the order of the Markov chain, otherwise it will overfit. In fact, this problem is now aggravated by the larger number of model parameters. Therefore, we employ a Bayesian formulation and construct a generative process for the model parameters themselves. We do this by introducing prior probability densities for the parameters $\mathcal{D}_r(\{\theta_x\}|\alpha)$ and $\mathcal{D}_s(\{\lambda_{rs}\}|\beta)$ for tokens and memories, respectively, with

**Table 1 Summary of inference results for empirical sequences**

| n | US flight itineraries | | | | War and Peace | | | | Taxi movements | | | | RockYou password list | | | |
|---|---|---|---|---|---|---|---|---|---|---|---|---|---|---|---|---|
| | $B_N$ | $B_M$ | $\Sigma$ | $\Sigma'$ | $B_N$ | $B_M$ | $\Sigma$ | $\Sigma'$ | $B_N$ | $B_M$ | $\Sigma$ | $\Sigma'$ | $B_N$ | $B_M$ | $\Sigma$ | $\Sigma'$ |
| 1 | 384 | 365 | 364,385,780 | 365,211,460 | 65 | 71 | 11,422,564 | 11,438,753 | 387 | 385 | 2,635,789 | 2,975,299 | 140 | 147 | 1,060,272,230 | 1,060,385,582 |
| 2 | 386 | 7605 | 319,851,871 | 326,511,545 | 62 | 435 | 9,175,833 | 9,370,379 | 397 | 1127 | 2,554,662 | 3,258,586 | 109 | 1597 | 984,697,401 | 987,185,890 |
| 3 | 183 | 2455 | 318,380,106 | 339,898,057 | 70 | 1366 | 7,609,366 | 8,493,211 | 393 | 1036 | 2,590,811 | 3,258,586 | 114 | 4703 | 910,330,062 | 930,926,370 |
| 4 | 292 | 1558 | 318,842,968 | 337,988,629 | 72 | 1150 | 7,574,332 | 9,282,611 | 397 | 1071 | 2,628,813 | 3,258,586 | 114 | 5856 | 889,006,060 | 940,991,463 |
| 5 | 297 | 1573 | 335,874,766 | 338,442,011 | 71 | 882 | 10,181,047 | 10,992,795 | 395 | 1095 | 2,664,990 | 3,258,586 | 99 | 6430 | 1,000,410,410 | 1,005,057,233 |
| gzip | | | 573,452,240 | | | | 9,594,000 | | | | 4,289,888 | | | | 1,315,388,208 | |
| LZMA | | | 402,125,144 | | | | 7,420,464 | | | | 2,902,904 | | | | 1,097,012,288 | |

Description length $\Sigma = -\log_2 P(\{x_t\}, b)$ in bits, as well as inferred number of token groups $B_N$ and memory groups $B_M$ for different data sets and Markov order $n$ (for detailed descriptions, see Methods section Data sets). The value $\Sigma' = -\log_2 P(\{x_t\})$ corresponds to the direct Bayesian parametrisation of Markov chains of ref. [28], with noninformative priors. Values in grey correspond to the minimum of each column. The bottom rows show the compression obtained with gzip and LZMA, two popular variations of Lempel-Ziv[54, 55].

hyperparameter sets $\alpha$ and $\beta$, and computing the integrated likelihood

$$P(\{x_t\}|\alpha, \beta, b) = \int d\theta d\lambda \, P(\{x_t\}|b, \lambda, \theta)$$
$$\times \prod_r \mathcal{D}_r(\{\theta_x\}|\alpha) \prod_s \mathcal{D}_s(\{\lambda_{rs}\}|\beta). \tag{8}$$

where we used $b$ as a shorthand for $\{b_x\}$ and $\{b_x\}$. Now, instead of inferring the hyperparameters, we can make a noninformative choice for $\alpha$ and $\beta$ that reflects our a priori lack of preference towards any particular model[30]. Doing so in this case yields a likelihood (for details, see Methods section Bayesian Markov chains with communities),

$$P(\{x_t\}|b, \{e_s\}) = P(\{x_t\}|b, \{e_{rs}\}, \{k_x\})$$
$$\times P(\{k_x\}|\{e_{rs}\}, b)P(\{e_{rs}\}|\{e_s\}), \tag{9}$$

where $P(\{x_t\}|b, \{e_{rs}\}, \{k_x\})$ corresponds to the likelihood of the sequence $\{x_t\}$ conditioned on the transitions counts $\{e_{rs}\}$ and token frequencies $\{k_x\}$, and the remaining terms are the prior probabilities on the discrete parameters $\{e_{rs}\}$ and $\{k_x\}$. Since the likelihood above still is conditioned on the partitions $\{b_x\}$ and $\{b_x\}$, as well as the memory group counts $\{e_s\}$, we need to include prior probabilities on these as well to make the approach fully nonparametric. Doing so yields a joint likelihood for both the sequence and the model parameters,

$$P(\{x_t\}, b, \{e_s\}) = P(\{x_t\}|b, \{e_s\})P(b)P(\{e_s\}). \tag{10}$$

It is now possible to understand why maximising this joint likelihood will prevent overfitting the data. If we take its negative logarithm, it can be written as

$$\Sigma = -\log_2 P(\{x_t\}, b, \{e_s\}) \tag{11}$$

$$= -\log_2 P(\{x_t\}|b, \{e_s\}) - \log_2 P(b, \{e_s\}). \tag{12}$$

The quantity $\Sigma$ is called the description length of the data[31, 32]. It corresponds to the amount of information necessary to describe both the data and the model simultaneously, corresponding to the first and second terms in eq. (12), respectively. As the model becomes more complex—either by increasing the number of groups or the order of the Markov chain—this will decrease the first term as the data likelihood increases, but it will simultaneously increase the second term, as the model likelihood decreases. The second term then acts as a penalty to the model likelihood, forcing a balance between model complexity and quality of fit. Unlike approximate penalty approaches based solely on the number of free parameters such as BIC[33] and AIC[34], which are not to valid for network models[35], the description length of the model is exact and fully captures its flexibility. Because of the complete character of the description length,

minimising it indeed amounts to achieving true compression of data, differently from the parametric maximum likelihood approach mentioned earlier. Because the whole process is functionally equivalent to inferring the SBM for networks, we can use the same algorithms[36] (for a details about the inference method, see Methods section 'Bayesian Markov chains with communities').

Before we continue, we point out that the selection of priors in eq. (9) needs to be done carefully to avoid underfitting the data. This happens when strong prior assumptions obscure structures in the data[37]. We tackle this by using hierarchical priors, where the parameter themselves are modelled by parametric distributions, which in turn contain more parameters, and so on[38, 39]. Besides alleviating the underfitting problem, this allows us to represent the data in multiple scales by a hierarchical partition of the token and memories. We describe this in more detail in the Methods section Bayesian Markov chains with communities.

This Markov chain model with communities succeeds in providing a better description for a variety of empirical sequences when compared with the common Markov chain parametrisation (see Table 1). Not only do we systematically observe a smaller description length, but we also find evidence for higher-order memory in all examples. We emphasise that we are protected against overfitting: If we randomly shuffle the order of the tokens in each data set, with dominating probability we infer a fully random model with $n = 1$ and $B_N = B_M = 1$, which is equivalent to an $n = 0$ memoryless model. We have verified that we infer this model for all analysed data sets. Accordingly, we are not susceptible to the spurious results of nonstatistical methods[23].

To illustrate the effects of community structure on the Markov dynamics, we use the US flight itineraries as an example. In this data set, the itineraries of 1,272,696 passengers were recorded, and we treat each airport stop as a token in a sequence (for more details, see Methods section Data sets). When we infer our model, the itinerary memories are grouped together if their destination probabilities are similar. As a result, it becomes possible, for example, to distinguish transit hubs from destination hubs[2]. We use Atlanta and Las Vegas to illustrate: Many roundtrip routes transit through Atlanta from the origin to the final destination and return to it two legs later on the way back to the origin. On the other hand, Las Vegas often is the final destination of a roundtrip such that the stop two legs later represents a more diverse set of origins (Fig. 3). Resembling the results of the map equation for network flows with memory[2], this pattern is captured in our model by the larger number of memory groups that involve Las Vegas than those that involve Atlanta. Moreover, the division between transit and destinations propagates all the way to the upper hierarchical levels of the memory partition.

In addition to this itinerary memory clustering, the co-clustering with airport tokens also divides the airports into hierarchical categories. For example, Atlanta is grouped with nearby Charlotte at the first hierarchy level, and with Detroit,

Minneapolis, Dallas and Chicago at the third level. This extra information tells us that these airports serve as alternative destinations to itineraries that are similar to those that go through Atlanta. Likewise, Las Vegas is grouped together with alternative destinations Phoenix and Denver. This type of similarity between airports—which is not merely a reflection of the memory patterns —is not expressed with the assortative approach of the map equation, which solely clusters densely connected memories with long flow persistence times[2]. A more direct comparison between our Bayesian inference framework and the map equation is not meaningful, since these two approaches represent the network divisions differently (for a detailed discussion, see Methods section 'Comparison with the map equation for network flows with memory'). Indeed, it is the simultaneous division of memories and tokens that effectively reduce the overall complexity of the data, and provide better compression at higher memory order. Consequently, the community-based Markov model can capture patterns of higher-order memory that conventional methods obscure.

**Temporal networks**. A general model for temporal networks treats the edge sequence as a time series[6, 40, 41]. We can in principle use the present model without any modification by considering the observed edges as tokens in the Markov chain, that is, $x_t = (i, j)_t$, where $i$ and $j$ are the endpoints of the edge at time $t$ (see Fig. 1b). However, this can be suboptimal if the networks are sparse, that is, if only a relatively small subset of all possible edges occur, and thus there are insufficient data to reliably fit the model. Therefore, we adapt the model above by including an additional generative layer between the Markov chain and the observed edges. We do so by partitioning the nodes of the network into groups, that is, $c_i \in [1, C]$ determines the membership of node $i$ in one of $C$ groups, such that each edge $(i, j)$ is associated with a label $(c_i, c_j)$. Then we define a Markov chain for the sequence of edge labels and sample the actual edges conditioned only on the labels. Since this reduces the number of possible tokens from $O(N^2)$ to $O(C^2)$, it has a more controllable number of parameters that can better match the sparsity of the data. We further assume that, given the node partitions, the edges themselves are sampled in a degree-corrected manner, conditioned on the edge labels,

$$P((i,j)|(r,s),\kappa,c) = \begin{cases} \delta_{c_i,r}\delta_{c_j,s}\kappa_i\kappa_j & \text{if } r \neq s \\ 2\delta_{c_i,r}\delta_{c_j,s}\kappa_i\kappa_j & \text{if } r = s \end{cases}, \quad (13)$$

where $\kappa_i$ is the probability of a node being selected inside a group, with $\sum_{i \in r} \kappa_i = 1$. The total likelihood conditioned on the label sequence becomes

$$P(\{(i,j)_t\}|\{(r,s)_t\},\kappa,c) = \prod_t P((i,j)_t|(r,s)_t,\kappa). \quad (14)$$

Since we want to avoid overfitting the model, we once more use noninformative priors, but this time on $\{\kappa_i\}$, and integrate over them,

$$P(\{(i,j)_t\}|\{(r,s)_t\},c)$$
$$= \int P(\{(i,j)_t\}|\{(r,s)_t\},\kappa,c)P(\kappa)\,\mathrm{d}\kappa. \quad (15)$$

Combining this result with eq. (9), we have the complete likelihood of the temporal network,

$$P(\{(i,j)_t\}|c,b) = P(\{(i,j)_t\}|\{(r,s)_t\},c)P(\{(r,s)_t\}|b), \quad (16)$$

conditioned only on the partitions. As we show in detail in the Methods section Temporal networks, this model is a direct generalisation of the static DCSBM, with a likelihood composed of two separate static and dynamic terms. One recovers the static DCSBM exactly by choosing $B_N = B_M = 1$, making the state transitions memoryless.

Finally, to make the model nonparametric, we again include the same prior as before for the node partition $c$, in addition to token and memory partition $b$, such that the total nonparametric joint likelihood is maximised,

$$P(\{(i,j)_t\},c,b) = P(\{(i,j)_t\}|c,b)P(c)P(b). \quad (17)$$

In this way, we again protect against overfitting, and we can infer not only the number of memory groups $B_N$ and token groups $B_M$, as before, but also the number of groups in the temporal network itself, $C$. If, for example, the temporal network is completely random—that is, the edges are placed randomly both in the aggregated network as well as in time—we again infer $B_N = B_M = C = 1$ with the largest probability. We refer to the Methods section Temporal networks for a complete derivation of the final likelihood.

We employ this model in a variety of dynamic network data sets from different domains (for details, see Table 2 and Methods section Data sets). In all cases, we infer models with $n > 0$ that identify many groups for the tokens and memories, meaning that the model succeeds in capturing temporal structures. In most cases, models with $n = 1$ best describe the data, implying that there is not sufficient evidence for higher-order memory, with exception of the network of chess moves, which is best described by a model with $n = 2$. This result is different from the results for the comparably long non-network sequences in Table 1, where we identified higher-order Markov chains. Again, this is because the alphabet size is much larger for temporal networks—corresponding to all possible edges that can be encountered. Hence, for the data sets in Table 2 the size of the alphabet is often comparable with the length of the sequence. In view of this, it is remarkable that the method can detect any structure at all. The intermediary layer where the Markov chain generates edge types instead of the edges directly is crucial. If we fit the original model without this modification, we indeed get much larger description lengths and we often fail to detect any Markov structure (not shown).

To illustrate how the model characterises the temporal structure of these systems, we focus on the proximity network of high school students, which corresponds to the voluntary tracking of 327 students for a period of 5 days[42]. Whenever the distance between two students fell below a threshold, an edge between them was recorded at that time. In the best-fitting model for these data, the inferred groups for the aggregated network correspond exactly to the known division into 9 classes, except for the PC class, which was divided into two groups (Fig. 4). The groups show a clear assortative structure, where most connections occur within each class. The clustering of the edge labels in the second part of the model reveals the temporal dynamics. We observe that the edges connecting nodes of the same group cluster either in single-node or small groups, with a high incidence of self-loops. This means that if an edge that connects two students of the same class appears in the sequence, the next edge is most likely also inside the same class, indicating that the students of the same class are clustered in space and time. The remaining edges between students of different classes are separated into two large groups. This division indicates that the different classes meet each other at different times. Indeed, the classes are located in different parts of the school building and they typically go to lunch separately[42]. Accordingly, our method can uncover the associated dynamical pattern from the data alone.

**Temporal prediction**. Using generative models to extract patterns in data yields more than a mere description, since they generalise observations and predict future events. For our Bayesian approach, the models can even be used to predict tokens and memories not previously observed. This ability is in strong contrast to more heuristic methods that are only designed to find partitions in networks or time series, and cannot be used for prediction. Furthermore, the predictive performance of a model is often used on its own to evaluate it, and serves as an alternative approach to model selection: since an overly complicated model incorporates noise in its description, it yields less accurate predictions. Thus, maximising the predictive performance also amounts to a balance between quality of fit and model complexity, similarly to the minimum description length approach we have used so far. It is important, therefore, to determine if these two different criteria yield consistent results, which would serve as an additional verification of the overall approach.

We show this consistency by considering a scenario where a sequence is divided into two equal-sized contiguous parts, $\{x_t\} = \{x_t^*\} \cup \{x_t'\}$. That is, a training set $\{x_t^*\}$ and a validation set $\{x_t'\}$. We then evaluate the model by fitting it to the training set and use it to predict the validation set. If we observe only the training set, the likelihood of the validation set is bounded below by $P(\{x_t'\}|\{x_t^*\}, b^*) \geq \exp(-\Delta\Sigma)$, where $b^* = \arg\max_b P(b|\{x_t^*\})$ is the best partition given the training set and $\Delta\Sigma$ is the description length difference between the training set and the entire data (for a proof, see Methods section Predictive held-out likelihood). This lower bound will become tight when both the validation and training sets become large, and hence can be used as an asymptotic approximation of the predictive likelihood. Table 2 shows empirical values for the same data sets as considered before, where $n = 0$ corresponds to using only the static DCSBM to predict the edges, ignoring any time structure. The temporal network model provides better prediction in all cases, and the best Markov order always coincides with the one that yields the minimum description length, thus confirming a full agreement between both criteria in these cases.

## Discussion

We presented a dynamical variation of the degree-corrected stochastic block model that can capture long pathways or large-scale structures in sequences and temporal networks. The model does not require the optimal Markov order or number of groups as inputs, but infers them from data because the underlying arbitrary-order Markov chain model is nonparametric. Its nonparametric nature also evades a priori imposed timescales. We showed that the model successfully finds meaningful large-scale temporal structures in real-world systems and that it predicts their temporal evolution. Moreover, in the Methods section we extend the model to situations where the dynamics take place in continuous time or is nonstationary. In contrast to approaches that force network-formation dynamics into discrete time windows, and require a priori knowledge about the appropriate amount of dynamical memory, our approach provides a principled and versatile alternative.

## Methods

**Bayesian Markov chains with communities**. As described in the main text, a Bayesian formulation of the Markov model consists in specifying prior probabilities for the model parameters, and integrating over them. In doing so, we convert the problem from one of parametric inference where the model parameters need to be specified before inference, to a nonparametric one where no parameters need to be specified before inference. In this way, the approach possesses intrinsic regularisation, where the order of the model can be inferred from data alone, without overfitting[30, 43].

To accomplish this, we rewrite the model likelihood, using eqs. (1) and (5), as

$$P(\{x_t\}|b, \lambda, \theta) = \prod_{x, x} (\theta_x \lambda_{b_x, b_x})^{a_{x,x}} = \prod_x \theta_x^{k_x} \prod_{r<s} \lambda_{rs}^{e_{rs}}, \tag{18}$$

and observe the normalisation constraints $\sum_{x \in r} \theta_x = 1$, and $\sum_s \lambda_{rs} = 1$. Since this is just a product of multinomials, we can choose conjugate Dirichlet priors probability densities $\mathcal{D}_r(\{\theta_x\}|\{\alpha_x\})$ and $\mathcal{D}_s(\{\lambda_{rs}\}|\{\beta_{rs}\})$, which allows us to exactly compute the integrated likelihood,

$$\begin{aligned} P(\{x_t\}|\alpha, \beta, b) &= \int d\theta d\lambda \, P(\{x_t\}|b, \lambda, \theta) \\ &\quad \times \prod_r \mathcal{D}_r(\{\theta_x\}|\{\alpha_x\}) \prod_s \mathcal{D}_s(\{\lambda_{rs}\}|\{\beta_{rs}\}) \\ &= \left[ \prod_r \frac{\Gamma(A_r)}{\Gamma(e_r + A_r)} \prod_{x \in r} \frac{\Gamma(k_x + \alpha_x)}{\Gamma(\alpha_x)} \right] \\ &\quad \times \left[ \prod_s \frac{\Gamma(B_s)}{\Gamma(e_s + B_s)} \prod_r \frac{\Gamma(e_{rs} + \beta_{rs})}{\Gamma(\beta_{rs})} \right], \end{aligned} \tag{19}$$

where $A_r = \sum_{x \in r} \alpha_x$ and $B_s = \sum_r \beta_{rs}$. We recover the Bayesian version of the common Markov chain formulation (see ref. [28]) if we put each memory and token in their own groups. This remains a parametric distribution, since we need to specify the hyperparameters. However, in the absence of prior information it is more appropriate to make a noninformative choice that encodes our a priori lack of knowledge or preference towards any particular model, which amounts to choosing $\alpha_x = \beta_{rs} = 1$, making the prior distributions flat. If we substitute these values in eq. (19), and re-arrange the terms, we can show that it can be written as the following combination of conditional likelihoods,

$$\begin{aligned} P(\{x_t\}|b, \{e_{rs}\}) &= P(\{x_t\}|b, \{e_{rs}\}, \{k_x\}) \\ &\quad \times P(\{k_x\}|\{e_{rs}\}, b) P(\{e_{rs}\}|\{e_s\}), \end{aligned} \tag{20}$$

where

$$P(\{x_t\}|b, \{e_{rs}\}, \{k_x\}) = \frac{\prod_{r<s} e_{rs}!}{\prod_r e_r! \prod_s e_s!} \prod_x k_x!, \tag{21}$$

$$P(\{k_x\}|\{e_{rs}\}, b) = \left[ \prod_r \left( \binom{n_r}{e_r} \right) \right]^{-1}, \tag{22}$$

$$P(\{e_{rs}\}|\{e_s\}) = \left[ \prod_s \left( \binom{B_N}{e_s} \right) \right]^{-1}, \tag{23}$$

with $\left( \binom{m}{n} \right) = \binom{m+n-1}{n}$ being the multiset coefficient, that counts the number of $m$-combinations with repetitions from a set of size $n$. The expression above has the following combinatorial interpretation: $P(\{x_t\}|b, \{e_{rs}\}, \{k_x\})$ corresponds to the likelihood of a microcanonical model[39] where a random

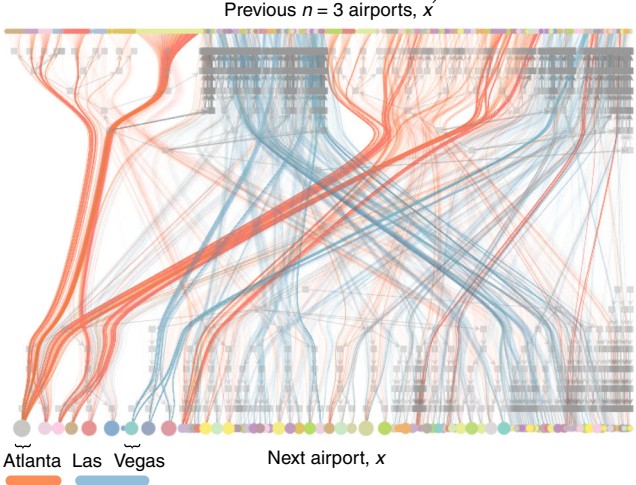

**Previous $n = 3$ airports, $\vec{x}$**

Atlanta  Las Vegas          **Next airport, $x$**

**Fig. 3** Selection of US flight itineraries for a third-order model. The itineraries contain stops in Atlanta or Las Vegas. Edges incident on memories of the type $\boldsymbol{x} = (x_{t-1}, \text{Atlanta}, x_{t-3})$ in *red* and $\boldsymbol{x} = (x_{t-1}, \text{LasVegas}, x_{t-3})$ in *blue*. The node colours and overlaid hierarchical division derive from the $n = 3$ model inferred for the whole dataset

sequence $\{x_t\}$ is produced with exactly $e_{rs}$ total transitions between groups $r$ and $s$, and with each token $x$ occurring exactly $k_x$ times. In order to see this, consider a chain where there are only $e_{rs}$ transitions in total between token group $r$ and memory group $s$, and each token $x$ occurs exactly $k_x$ times. For the first transition in the chain, from a memory $\boldsymbol{x}_0$ in group $s$ to a token $x_1$ in group $r$, we have the probability

$$P(x_1|\boldsymbol{x}_0, b, \{e_{rs}\}, \{k_x\}) = \frac{e_{rs}k_{x_1}}{e_s e_r}. \tag{24}$$

Now, for the second transition from memory $\boldsymbol{x}_1$ in group $t$ to a token $x_2$ in group $u$, we have the probability

$$P(x_2|\boldsymbol{x}_1, b, \{e_{rs}\}, \{k_x\}) =$$

$$\begin{cases} \frac{e_{ut}k_{x_2}}{e_t e_u}, & \text{if } t\neq s, \ u\neq r, \ x_2 \neq x_1, \\[2mm] \frac{(e_{us}-1)k_{x_2}}{(e_s-1)e_u}, & \text{if } t = s, \ u\neq r, \ x_2 \neq x_1, \\[2mm] \frac{e_{rt}(k_{x_1}-1)}{e_t(e_r-1)}, & \text{if } t\neq s, \ u = r, \ x_2 = x_1, \\[2mm] \frac{e_{rt}k_{x_2}}{e_t(e_r-1)}, & \text{if } t\neq s, \ u = r, \ x_2 \neq x_1, \\[2mm] \frac{(e_{rs}-1)k_{x_2}}{(e_s-1)(e_r-1)}, & \text{if } t = s, \ u = r, \ x_2 \neq x_1, \\[2mm] \frac{(e_{rs}-1)(k_{x_1}-1)}{(e_s-1)(e_r-1)}, & \text{if } t = s, \ u = r, \ x_2 = x_1. \end{cases} \tag{25}$$

Proceeding recursively, the final likelihood for the entire chain is

$$P(\{x_t\}|b, \{e_{rs}\}, \{k_x\}) = \frac{\prod_{rs} e_{rs}!}{\prod_r e_r! \prod_s e_s!} \prod_x k_x!, \tag{26}$$

which is identical to eq. (21).

The remaining terms in eqs. (22) and (23) are the prior probabilities on the discrete parameters $\{k_x\}$ and $\{e_{rs}\}$, respectively, which are uniform distributions of the type $1/\Omega$, where $\Omega$ is the total number of possibilities given the imposed constraints. We refer to ref. [39] for a more detailed discussion on those priors.

Since the integrated likelihood above gives $P(\{x_t\}|b, \{e_s\})$, we still need to include priors for the node partitions $\{b_i\}$ and $\{b_x\}$, as well as memory group counts, $\{e_s\}$, to make the above model fully nonparametric. This is exactly the same situation encountered with the SBM[37–39]. Following refs. [38, 39], we use a nonparametric two-level Bayesian hierarchy for the partitions, $P(\{b_i\}) = P(\{b_i\}|\{n_r\})P(\{n_r\})$, with uniform distributions

$$P(\{b_i\}|\{n_r\}) = \frac{\prod_r n_r!}{M!}, \quad P(\{n_r\}) = \binom{M-1}{B-1}^{-1}, \tag{27}$$

where $n_r$ is the number of nodes in group $r$, $M = \sum_r n_r$, which we use for both $\{b_x\}$ and $\{b_i\}$, that is, $P(b) = P(\{b_x\})P(\{b_i\})$. Analogously, for $\{e_s\}$ we can use a uniform distribution

$$P(\{e_s\}|b) = \left( \binom{B_M}{E} \right)^{-1}. \tag{28}$$

The above priors make the model fully nonparametric with a joint and marginal probability $P(\{x_t\}, b) = P(\{x_t\}, b, \{e_s\}) = P(\{x_t\}|b, \{e_s\})P(b)P(\{e_s\})$. This approach successfully finds the most appropriate number of groups according to statistical evidence, without overfitting[32, 37, 38, 44]. This nonparametric method can also detect the most appropriate order of the Markov chain, again without overfitting[28]. However, in some ways it is still sub-optimal. The use of conjugate Dirichlet priors above was primarily for mathematical convenience, not because they closely represent the actual mechanisms believed to generate the data. Although the noninformative choice of the Dirichlet distribution (which yields flat priors for $\{e_{rs}\}$ and $\{e_s\}$) can be well justified by maximum entropy arguments (see ref. [30]), and are unbiased, it can in fact be shown that it can lead to underfitting of the data, where the maximum number of detectable groups scales sub-optimally as $\sqrt{N}$[37]. As shown in ref. [38], this limitation can be overcome by departing from the model with Dirichlet priors, and replacing directly the priors $P(\{e_{rs}\}|\{e_s\})$ and $P(\{e_s\})$ of the microcanonical model by a single prior $P(\{e_{rs}\})$, and noticing that $\{e_{rs}\}$ corresponds to the adjacency matrix of bipartite multigraph with $E$ edges and $B_N + B_M$ nodes. With this insight, we can write $P(\{e_{rs}\})$ as a Bayesian hierarchy of nested SBMs, which replaces the resolution limit above by $N/\ln N$, and provides a multilevel description of the data, while remaining unbiased. Furthermore, the uniform prior in eq. (8) for the token frequencies $P(\{k_x\}|\{e_{rs}\}, (b)$ intrinsically favours concentrated distributions of $k_x$ values. This distribution

is often skewed. We therefore replace it by a two-level Bayesian hierarchy $P(\{k_x\}|\{e_{rs}\}, b) = \prod_r P(\{k_x\}|\{n_k^r\})P(\{n_k^r\}|e_r)$, with

$$P(\{k_x\}|\{n_k^r\}) = \frac{\prod_k n_k^r!}{n_r!}, \tag{29}$$

and $P(\{n_k^r\}|e_r) = q(e_r, n_r)^{-1}$, where $q(m, n)$ is the number of restricted partitions of integer $m$ into at most $n$ parts (see ref. [39] for details).

As mentioned in the main text, in order to fit the model above we need to find the partitions $\{b_x\}$ and $\{b_x\}$ that maximise $P(\{x_t\}, b)$, or fully equivalently, minimise the description length $\Sigma = -\ln P(\{x_t\}, b)$[31]. Since this is functionally equivalent to inferring the DCSBM in networks, we can use the same algorithms. In this work we employed the fast multilevel MCMC method of ref. [36], which has log-linear complexity $O(N \log^2 N)$, where $N$ is the number of nodes (in our case, memories and tokens), independent of the number of groups.

**The unified first-order model**. The model defined in the main text is based on a co-clustering of memory and tokens. In the $n = 1$ case, each memory corresponds to a single token. In this situation, we consider a slight variation of the model where we force the number of groups of each type to be the same, that is, $B_N = B_M = B$, and both partitions to be identical. Instead of clustering the original bipartite graph, this is analogous to clustering its projection into a directed transition graph with each node representing a specific memory and token simultaneously. When considering this model, the likelihoods computed in the main text and above remain exactly the same, with the only difference that we implicitly force both memory and token partitions to be identical, and omit the partition likelihood of eq. (27) for one of them. We find that for many data sets this variation provides a slightly better description than the co-clustering version, although there are also exceptions to this.

We used this variation of the model in Fig. 4 because it yielded a smaller description length for that dataset, and simplified the visualisation and interpretation of the results in that particular case.

**Temporal networks**. Here we show in more detail how the likelihood for the temporal network model is obtained. As we discuss in the Results section Temporal networks, the total likelihood of the network conditioned on the label sequence is

$$P(\{(i,j)_t\}|\{(r,s)_t\}, \kappa, c) = \prod_i P((i,j)_t|(r,s)_t, \kappa)$$
$$= \left[ \prod_t \delta_{c_{i_t}, r_t} \delta_{c_{j_t}, s_t} \right] \prod_i \kappa_i^{d_i} \prod_r 2^{m_{rr}}, \tag{30}$$

where $d_i$ is the degree of node $i$, and $m_{rs}$ is the total number of edges between groups $r$ and $s$. Maximum likelihood estimation gives $\hat{\kappa}_i = d_i/e_{c_i}$. But since we want to avoid overfitting the model, we once more use noninformative priors, this time on $\{\kappa_i\}$, integrate over them, henceforth omitting the trivial Kronecker delta term above and obtain

$$P(\{(i,j)_t\}|\{(r,s)_t\}, c) = \frac{\prod_i d_i! \prod_r 2^{m_{rr}}}{\prod_r e_r!} P(\{d_i\}), \tag{31}$$

with $P(\{d_i\}) = \prod_r \left( \binom{n_r}{e_r} \right)^{-1}$. Combining this with eq. (9) as $P(\{(i,j)_t\}|c, b) = P(\{(i,j)_t\}|\{(r,s)_t\}, c)P(\{(r, s)_t\}|b)$, we have the complete likelihood of the temporal network

$$P(\{(i,j)_t\}|c, b) = \frac{\prod_{r \geq s} m_{rs}! \prod_r 2^{m_{rr}}}{\prod_r e_r!} \prod_i d_i! \tag{32}$$

$$\times P(\{d_i\}|c)P(\{m_{rs}\}) \frac{\prod_{u<v} e'_{uv}!}{\prod_u e'_{u}! \prod_v e'_{v}!} P(\{e'_{uv}\}). \tag{33}$$

This likelihood can be rewritten in such a way that makes clear that it is composed of one purely static and one purely dynamic part,

$$P(\{(i,j)_t\}|c, b) = P(\{A_{ij}\}|c) \times \frac{P(\{(r,s)_t\}|b, \{e_v\})}{P(\{m_{rs}\}) \prod_{r \geq s} m_{rs}!}. \tag{34}$$

The first term of eq. (34) is precisely the nonparametric likelihood of the static DCSBM that generates the aggregated graph with adjacency matrix $A_{ij} = k_x = (i,j)$ given the node partition $\{c_i\}$, which itself is given by

$$\ln P(\{A_{ij}\}|c) \approx E + \frac{1}{2} \sum_{rs} e_{rs} \ln \frac{e_{rs}}{e_r e_s} + \sum_i \ln d_i! $$
$$+ \ln P(\{d_i\}) + \ln P(\{m_{rs}\}), \tag{35}$$

if Stirling's approximation is used. The second term in eq. (34) is the likelihood of the Markov chain of edge labels given by eq. (9) (with $\{x_t\} = \{(r,s)_t\}$, and $\{k_x\} = \{m_{rs}\}$). This model, therefore, is a direct generalisation of the static DCSBM, with a likelihood composed of two separate static and dynamic terms. One recovers the static DCSBM exactly by choosing $B_N = B_M = 1$—making the state transitions

**Table 2 Summary of inference results for empirical temporal networks**

| | High school proximity (N = 327, E = 5818) | | | | | Enron email (N = 87,273, E = 1,148,072) | | | | | Internet AS (N = 53,387, E = 500,106) | | | | |
|---|---|---|---|---|---|---|---|---|---|---|---|---|---|---|---|
| n | C | $B_N$ | $B_M$ | Σ | −ΔΣ | C | $B_N$ | $B_M$ | Σ | −ΔΣ | C | $B_N$ | $B_M$ | Σ | −ΔΣ |
| 0 | 10 | - | - | 89,577 | −64,129 | 1447 | - | - | 19,701,405 | −11,631,987 | 187 | - | - | 19,701,403 | −8,094,541 |
| 1 | 10 | 9 | 9 | 82,635 | −49,216 | 1596 | 2219 | 2201 | 13,107,399 | −8,012,378 | 185 | 131 | 131 | 10,589,136 | −6,279,923 |
| 2 | 10 | 6 | 6 | 86,249 | −49,533 | 324 | 366 | 313 | 16,247,904 | −8,370,876 | 132 | 75 | 43 | 14,199,548 | −6,921,032 |
| 3 | 9 | 6 | 6 | 103,453 | −49,746 | 363 | 333 | 289 | 26,230,928 | −14,197,057 | 180 | 87 | 79 | 22,821,016 | −8,133,665 |
| | **APS citations (N = 425,760, E = 4,262,443)** | | | | | **prosper.com loans (N = 89,269, E = 3,394,979)** | | | | | **Chess moves (N = 76, E = 3,130,166)** | | | | |
| 0 | 3774 | - | - | 131,931,579 | −93,802,176 | 318 | - | - | 96,200,002 | −64,428,332 | 72 | - | - | 66,172,128 | −34,193,040 |
| 1 | 4426 | 6853 | 6982 | 94,523,280 | −56,059,700 | 267 | 1039 | 1041 | 59,787,374 | −30,487,941 | 72 | 339 | 339 | 58,350,128 | −30,271,323 |
| 2 | 4268 | 710 | 631 | 144,887,083 | −100,264,678 | 205 | 619 | 367 | 109,041,487 | −54,211,919 | 72 | 230 | 266 | 58,073,342 | −30,110,657 |
| 3 | 4268 | 454 | 332 | 228,379,667 | −120,180,052 | 260 | 273 | 165 | 175,269,743 | −54,655,474 | 72 | 200 | 205 | 76,465,862 | −32,120,845 |
| | **Hospital contacts (N = 75, E = 32,424)** | | | | | **Infectious Sociopatterns (N = 10,972, E = 415,912)** | | | | | **Reality Mining (N = 96, E = 1,086,404)** | | | | |
| 0 | 68 | - | - | 484,121 | −270,355 | 4695 | - | - | 8,253,351 | −6,876,439 | 93 | - | - | 21,337,812 | −10,835,792 |
| 1 | 60 | 58 | 58 | 245,479 | −131,010 | 5572 | 2084 | 2084 | 4,525,629 | −5,834,112 | 93 | 1015 | 1015 | 14,592,018 | −7,813,217 |
| 2 | 62 | 29 | 26 | 366,351 | −201,047 | 5431 | 3947 | 3947 | 7,503,859 | −6,311,244 | 95 | 1094 | 2541 | 14,657,975 | −8,185,791 |
| 3 | 50 | 11 | 7 | 644,083 | −332,889 | 1899 | 829 | 783 | 12,527,730 | −9,776,214 | 92 | 1225 | 1896 | 16,482,714 | −8,669,765 |

Description length $\Sigma = -\log_2 P(\{(i,j)_t\}, c, b)$ in bits as well as inferred number of node groups C, token groups $B_N$, and memory groups $B_M$ for different data sets and different Markov order n (see Methods section Datasets). The value $-\Delta\Sigma \leq \ln P(\{x'_t\}|\{x^*_t\}, b^*)$ is a lower-bound on the predictive likelihood of the validation set $\{x'_t\}$ given the training set $\{x^*_t\}$ and its best parameter estimate. Values in grey correspond to the minimum of each column

memoryless—so that the second term in eq. (34) above contributes only with a trivial constant $1/E!$ to the overall likelihood. Equivalently, we can view the DCSBM as a special case with $n = 0$ of this temporal network model.

**Predictive held-out likelihood.** Given a sequence divided in two contiguous parts, $\{x_t\} = \{x^*_t\} \cup \{x'_t\}$, that is, a training set $\{x : t^*\}$ and a validation set $\{x'_t\}$, and if we observe only the training set, the predictive likelihood of the validation set is

$$P(\{x'_t\}|\{x^*_t\}, b^*) = \frac{P(\{x'_t\} \cup \{x^*_t\}|b^*)}{P(\{x^*_t\}|b^*)}, \quad (36)$$

where $b^* = \text{argmax}_b P(b|\{x^*_t\})$ is the best partition given the training set. Moreover, we have

$$P(\{x'_t\} \cup \{x^*_t\}|b)^* = \sum_{b'} P(\{x'_t\} \cup \{x^*_t\}|b^*, b')P(b'|b^*), \quad (37)$$

where $b'$ corresponds to the partition of the newly observed memories (or even tokens) in $\{x'_t\}$. Generally we have $P(b'|b^*) = P(b', b^*)/P(b^*)$, so that

$$\begin{aligned} P(\{x'_t\}|\{x^*_t\}, b^*) &= \frac{\sum_{b'} P(\{x'_t\} \cup \{x^*_t\}|b^*, b')P(b^*, b')}{P(\{x^*_t\}|b^*)P(b^*)} \\ &\geq \frac{P(\{x'_t\} \cup \{x^*_t\}|b^*, \hat{b}')P(b^*, \hat{b}')}{P(\{x^*_t\}|b^*)P(b^*)} = \exp(-\Delta\Sigma), \end{aligned} \quad (38)$$

where $\hat{b}' = \text{argmax}_{b'} P(\{x'_t\} \cup \{x^*_t\}|b^*, b')P(b^*, b')$ and $\Delta\Sigma$ is the difference in the description length between the training set and the entire data. Hence, computing the minimum description length of the remaining data by maximising the posterior likelihood relative to the partition of the previously unobserved memories or tokens, yields a lower bound on the predictive likelihood. This lower bound will become tight when both the validation and training sets become large, because then the posterior distributions concentrate around the maximum, and hence can be used as an asymptotic approximation of the predictive likelihood.

**Continuous time.** So far, we have considered sequences and temporal networks that evolve discretely in time. Although this is the appropriate description for many types of data, such as text, flight itineraries and chess moves, in many other cases events happen instead in real time. In this case, the time series can be represented—without any loss of generality—by an embedded sequence of tokens $\{x_t\}$ placed in discrete time, together with an additional sequence of waiting times $\{\Delta_t\}$, where $\Delta_t \geq 0$ is the real time difference between tokens $x_t$ and $x_{t-1}$. Employing a continuous-time Markov chain description, the data likelihood can be written as

$$P(\{x_t\}, \{\Delta_t\}|p, \lambda) = P(\{x_t\}|p) \times P(\{\Delta_t\}|\{x_t\}, \lambda) \quad (39)$$

with $P(\{x_t\}|p)$ given by eq. (1), and

$$P(\{\Delta_t\}|\{x_t\}, \lambda) = \prod_i P(\Delta_t|\lambda_{\mathbf{x}_{t-1}}), \quad (40)$$

where

$$P(\Delta|\lambda) = \lambda e^{-\lambda\Delta}, \quad (41)$$

is a maximum-entropy distribution governing the waiting times, according to the

frequency $\lambda$. Substituting this in eq. (40), we have

$$P(\{\Delta_t\}|\{x_t\}, \lambda) = \prod_x \lambda_x^{k_x} e^{-\lambda_x \Delta_x}, \quad (42)$$

where $\Delta_x = \sum_t \Delta_t \delta_{x_t, x}$. To compute the nonparametric Bayesian evidence, we need a conjugate prior for the frequencies $\lambda_x$,

$$P(\lambda|\alpha, \beta) = \frac{\beta^\alpha \lambda^{\alpha-1}}{\Gamma(\alpha)} e^{-\beta\lambda}, \quad (43)$$

where $\alpha$ and $\beta$ are hyperparameters, interpreted, respectively, as the number and sum of prior observations. A fully noninformative choice would entail $\alpha \to 0$ and $\beta \to 0$, which would yield the so-called Jeffreys prior[46], $P(\lambda) \propto 1/\lambda$. Unfortunately, this prior is improper because it is not a normalised distribution. In order to avoid this, we use instead $\alpha = 1$ and $\beta = \sum_x \lambda_x/M$, taking into account the global average. While this is not the only possible choice, the results should not be sensitive to this prior since the data will eventually override any reasonable assumption we make. Using this prior, we obtain the Bayesian evidence for the waiting times as

$$P(\{\Delta_t\}|\{x_t\}) = \prod_x \int_0^\infty \lambda^{k_x} e^{-\lambda\Delta_x} P(\lambda|\alpha, \beta) d\lambda, \quad (44)$$

$$= \prod_x \frac{\beta^\alpha \Gamma(k_x + \alpha)}{\Gamma(\alpha)(\Delta_x + \beta)^{k_x+\alpha}}. \quad (45)$$

Hence, if we employ the Bayesian parametrisation with communities for the discrete embedded model as we did previously, we have

$$P(\{x_t\}, \{\Delta_t\}, b) = P(\{x_t\}, b) \times P(\{\Delta_t\}|\{x_t\}), \quad (46)$$

with $P(\{x_t\}, b)$ given by eq. (10).

Since the partition of memories and tokens only influences the first term of eq. (46), corresponding to the embedded discrete-time Markov chain, $P(\{x_t\}, b)$, the outcome of the inference for any particular Markov order will not take into account the distribution of waiting times—although the preferred Markov order might be influenced by it. We can change this by modifying the model above, assuming that the waiting times are conditioned on the group membership of the memories,

$$\lambda_x = \eta_{b_x}, \quad (47)$$

where $\eta_r$ is a frequency associated with memory group $r$. The Bayesian evidence is computed in the same manner, integrating over $\eta_r$ with the noninformative prior of eq. (43), yielding

$$P(\{\Delta_t\}|\{x_t\}) = \prod_r \frac{\beta^\alpha \Gamma(e_r + \alpha)}{\Gamma(\alpha)(\Delta_r + \beta)^{e_r+\alpha}}, \quad (48)$$

where $\Delta_r = \sum_t \Delta_t \delta_{b_{x_t}, r}$. Since this assumes that the waiting times will be sampled from the same distribution inside each group, the inference procedure will take the waiting time into account, and will place memories with significantly different delays into different groups.

As an example of the use of this model variation, we consider a piano reduction of Beethoven's fifth symphony (extracted in MIDI format from the Mutopia project at http://www.mutopiaproject.org), represented as a sequence of $E = 4223$ notes of

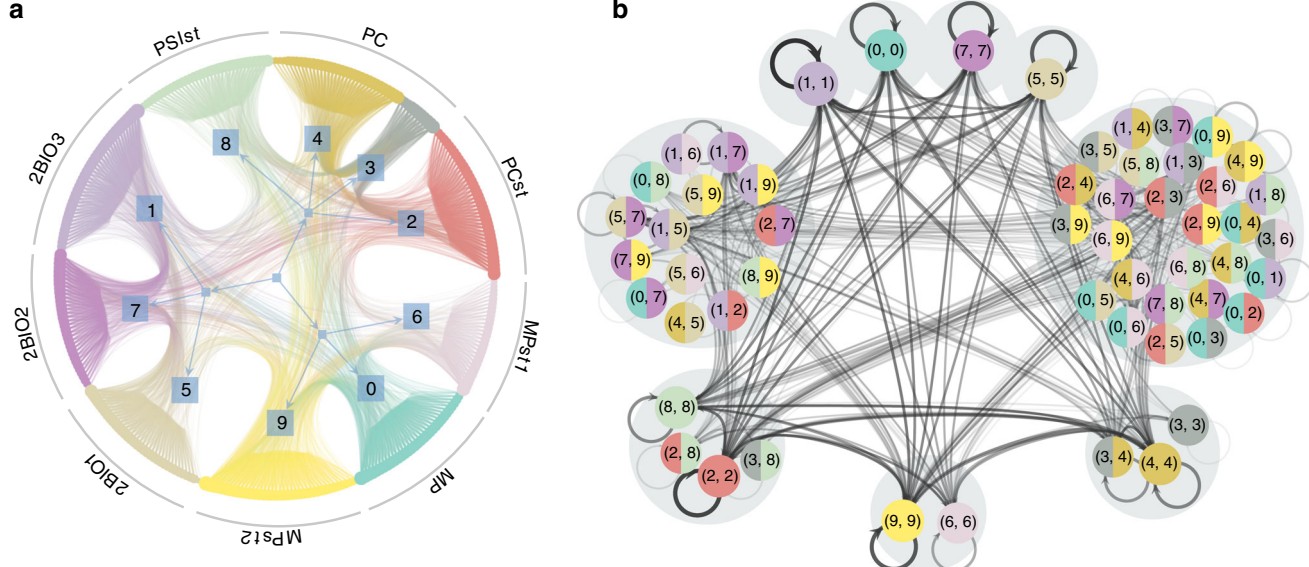

**Fig. 4** Inferred temporal model for a high school proximity network[42]. **a** The static part of the model divides the high school students into $C = 10$ groups (*square nodes*) that almost match the known classes (*text labels*). **b** The dynamic part of the model divides the directed multigraph group pairs in **a** into $B_N = B_M = 9$ groups (*grey circles*). The model corresponds to an $n = 1$ unified Markov chain on the edge labels, where the memory and tokens have identical partitions, as described in detail in the Methods section The unified first-order model

an alphabet of size $N = 63$. We consider both model variants, where the timings between notes are discarded, and where they are included. If individual notes are played simultaneously as part of a chord, we order them lexicographically and separate them by $\Delta t = 10^{-6}$ s. The results of the inference can be seen in Table 3. The discrete-time model favours an $n = 1$ Markov chain, whereas the continuous-time model favours $n = 2$. This is an interesting result that shows that the timings alone can influence the most appropriate Markov order. We can see in more detail why by inspecting the typical waiting times conditioned on the memory groups, as shown in Fig. 5. For the discrete-time model, the actual continuous waiting times (which are not used during inference) are only weakly correlated with the memory groups. On the other hand, for the continuous-time model we find that the memories are divided in such a way that they are strongly correlated with the waiting times. There is a group of memories for which the ensuing waiting times are always $\Delta t = 10^{-6}$, corresponding to node combinations that are always associated with chords. The remaining memories are divided into further groups that display at least two distinct timescales, that is, short and long pauses between notes. These statistically significant patterns are only visible for the higher order $n = 2$ model.

In the above model the waiting times are distributed according to the exponential distribution of eq. (41), which has a typical timescale given by $1/\lambda$. However, one often encounters processes where the dynamics are bursty, that is, the waiting times between events lack any characteristic scale, and are thus distributed according to a power-law

$$P(\Delta|\beta) = \frac{\beta \Delta_m^\beta}{\Delta^{\beta+1}}, \qquad (49)$$

for $\Delta > \Delta_m$, otherwise $P(\Delta|\beta) = 0$. One could in principle repeat the above calculations with the above distribution to obtain the inference procedure for this alternative model. However, this is in fact not necessary, since by making the transformation of variables

$$\mu = \ln \frac{\Delta}{\Delta_m}, \qquad (50)$$

we obtain for eq. (49)

$$P(\mu|\beta) = \beta e^{-\beta\mu}, \qquad (51)$$

which is the same exponential distribution of eq. (41). Hence, we need only to perform the transformation of eq. (50) for the waiting times prior to inference, to use the bursty model variant, while maintaining the exact same algorithm.

**Nonstationarity and hidden contexts.** An underlying assumption of the Markov model proposed is that the same transition probabilities are used for the whole duration of the sequence, that is, the Markov chain is stationary. Generalisations of the model can be considered where these probabilities change over time. Perhaps the simplest generalisation is to assume that the dynamics is divided into $T$ discrete

epochs, such that one replaces tokens $x_t$ by a pair $(x, \tau)_t$, where $\tau \in [1, T]$ represents the epoch where token $x$ was observed. In fact, $\tau$ does not need to be associated with a temporal variable—it could be any arbitrary covariate that describes additional aspects of the data. By incorporating this type of annotation into the tokens, one can use a stationary Markov chain describing the augmented tokens that in fact corresponds to a nonstationary one if one omits the variable $\tau$ from the token descriptors—effectively allowing for arbitrary extensions of the model by simply incorporating appropriate covariates, and without requiring any modification to the inference algorithm.

Another consequence of this extension is that the same token $x$ can belong to different groups if it is associated with two or more different covariates, $(x, \tau_1)$ and $(x, \tau_2)$. Therefore, this inherently models a situation where the group membership of tokens and memory vary in time.

As an illustration of this application of the model, we consider two literary texts: an English translation of 'War and peace', by Leo Tolstoy, and the French original of 'À la recherche du temps perdu', by Marcel Proust. First, we concatenate both novels together, treating it as a single text. If we fit our Markov model to it, we obtain the $n = 3$ model shown in Fig. 6a. In that figure, we have highlighted tokens and memories that involve letters that are exclusive to the French language, and thus most of them belong to the second novel. We observe that the model essentially finds a mixture between English and French. If, however, we indicate in each token to which novel it belongs, for example $(x, \text{wp})_t$ and $(x, \text{temps})_t$, we obtain the model of Fig. 6b. In this case, the model is forced to separate between the two novels, and one indeed learns the French patterns differently from English. Since this nonstationary model possesses a larger number of memory and tokens, one would expect a larger description length. However, in this cases it has a smaller description length than the mixed alternative, indicating indeed that both patterns are sufficiently different to warrant a separate description. Therefore, this approach is capable of uncovering change points[47], where the rules governing the dynamics change significantly from one period to another.

The above extension can also be used to uncover other types of hidden contexts. For example, in a temporal network of student proximity, we know that pairs of individuals that are far away are unlikely to be conditionally dependent on each other. If this spatial information is not available in the data, it may be inferred in same way it was done for language above. If the information is available, it can be annotated on the transitions, yielding a multilayer version of the model, similar to the layered SBM[15].

**Equivalence between structure and dynamics.** The likelihood of eq. (4) in the main text is almost the same as the DCSBM[29]. The only exceptions are trivial additive and multiplicative constants, as well as the fact that the degrees of the memories do not appear in it. These differences, however, do not alter the position of the maximum with the respect to the node partition. This allows us to establish an equivalence between inferring the community structure of networks and modelling the dynamics taking place on it. Namely, for a random walk on a connected undirected graph, a transition $i \rightarrow j$ is observed with probability $A_{ij}p_i(t)/$

**Table 3 Joint likelihoods for discrete- and continuous-time Markov models**

| | Discrete time | | | Continuous time | | |
|---|---|---|---|---|---|---|
| $n$ | $B_N$ | $B_M$ | $-\ln P(\{x_t\}, b)$ | $B_N$ | $B_M$ | $-\ln P(\{x_t\}, \{\Delta_t\}, b)$ |
| 1 | 40 | 40 | 13,736 | 37 | 37 | 58,128 |
| 2 | 35 | 34 | 15,768 | 24 | 22 | 47,408 |
| 3 | 34 | 33 | 24,877 | 16 | 15 | 54,937 |

Results inferred from Beethoven's fifth symphony for different Markov orders $n$. Values in grey correspond to the maximum likelihood of each column

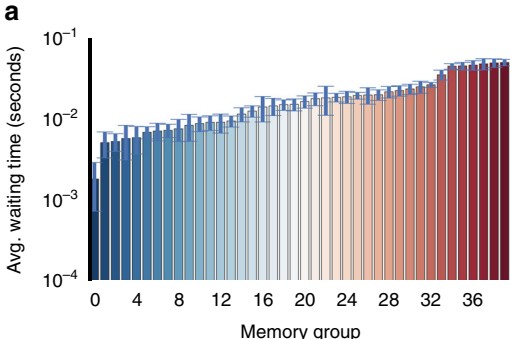

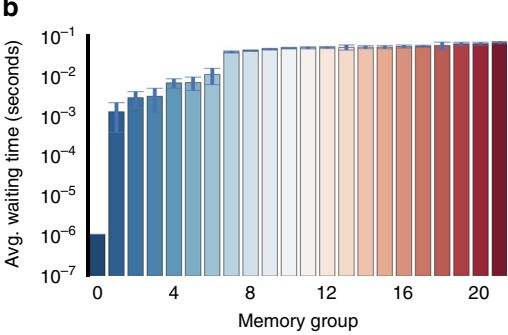

**Fig. 5** Waiting times for a discrete- and a continuous-time Markov model. Results inferred from Beethoven's fifth symphony. **a** $n = 1$ discrete-time model, ignoring the waiting times between notes. **b** $n = 2$ continuous-time model, with waiting times incorporated into the inference. The error bars correspond to the standard deviation of the mean

$k_i$, with $p_i(t)$ being the occupation probability of node $i$ at time $t$. Thus, after equilibrium with $p_i(\infty) = k_i/2E$, the probability of observing any edge $(i,j)$ is a constant: $p_i(\infty)/k_i + p_j(\infty)/k_j = 1/E$. Hence, the expected edge counts $e_{rs}$ between two groups in the Markov chain will be proportional to the actual edge counts in the underlying graph given the same node partition. This means that the likelihood of eq. (4) in the main text (for the $n = 1$ projected model described above) and of the DCSBM will differ only in trivial multiplicative and additive constants, such that the node partition that maximises them will be identical. This is similar to the equivalence between network modularity and random walks[48], but here the equivalence is stronger and we are not constrained to purely assortative modules. However, this equivalence breaks down for directed graphs, higher-order memory with $n > 1$ and when model selection chooses the number of groups.

**Comparison with the map equation for network flows with memory.** Both the community-based Markov model introduced here and the map equation for network flows with memory[2] identify communities in higher-order Markov chains based on maximum compression. However, the two approaches differ from each other in some central aspects. The approach presented here is based on the Bayesian formulation of a generative model, whereas the map equation finds a minimal modular entropy encoding of the observed dynamics projected on a node partition. Thus, both approaches seek compression, but of different aspects of the data.

The map equation operates on the internal and external transitions within and between possibly nested groups of memory states and describes the transitions

between physical nodes [$x_t$ is the physical node or token in memory states of the form $\mathbf{x} = (x_t, x_{t-1}, x_{t-2}, \ldots)$]. The description length of these transitions is minimised for the optimal division of the network into communities. By construction, this approach identifies assortative modules of memory states with long flow persistence times. Moreover, for inferring the most appropriate Markov order, this dynamics approach requires supervised approaches to model selection that uses random subsets of the data such as bootstrapping or cross validation[49].

On the other hand, the model presented here yields a nonparametric log-likelihood for the entire sequence as well as the model parameters, with its negative value corresponding to a description length for the entire data, not only its projection into groups. Minimising this description length yields the optimal co-clustering of memories and tokens, and hence assumes no inherent assortativity. Therefore it can be used also when the underlying Markov chain is disassortative. Moreover, the description length can also be used for unsupervised model selection, where the Markov order and number of groups are determined from the entire data, obviating the need for bootstrapping or cross validation. Furthermore, the present approach can be used to generate new data and make predictions based on past observations.

These distinctions mean that the two different approaches can give different results and that the problem at hand should decide which method to use.

**Data sets.** Below we give a description of the data sets used in this work.
US flight itineraries: This data set corresponds to a sample of flight itineraries in the US during 2011 collected by Bureau of Transportation Statistics of the United States Department of Transportation (http://www.transtats.bts.gov/). The data set contains 1,272,696 itineraries of varied lengths (airport stops). We aggregate all itineraries into a single sequence by concatenating the individual itineraries with a special separator token that marks the end of a single itinerary. There are 464 airports in the data set, and hence we have an alphabet of $N = 465$ tokens, and a single sequence with a total length of 83,653,994 tokens.
War and peace: This data set corresponds to the entire text of the english translation of the novel War and Peace by Leo Tolstoy, made available by the Project Gutenberg (extracted verbatim from https://www.gutenberg.org/cache/epub/2600/pg2600.txt). This corresponds to a sequence with an alphabet of size $N = 84$ (including letters, space, punctuation and special symbols) and a total length of 3,226,652 tokens.
Taxi movements: This data set contains GPS logs from 25,000 taxi pickups in San Francisco, collected by the company Uber (retrieved from http://www.infochimps.com/datasets/uber-anonymized-gps-logs, also available at https://github.com/dima42/uber-gps-analysis). The geographical locations were discretised into 416 hexagonal cells (see ref. [2] for details), and the taxi rides were concatenated together in a single sequence with a special separator token indicating the termination of a ride. In total, the sequence has an alphabet of $N = 417$ and a length of 819,172 tokens.
RockYou password list: This data set corresponds to a widely distributed list of 32,603,388 passwords from the RockYou video game company (retrieved from http://downloads.skullsecurity.org/passwords/rockyou-withcount.txt.bz2). The passwords were concatenated in a single sequence, with a special separator token between passwords. This yields a sequence with an alphabet of size $N = 215$ (letters, numbers and symbols) and a total length of 289,836,299 tokens.
High school proximity: This data set corresponds to a temporal proximity measurement of students in a french high school[42]. A total of $N = 327$ students were voluntarily tracked for a period of 5 days, generating $E = 5818$ proximity events between pairs of students.
Enron email: This data set corresponds to time-stamped collection of $E = 1,148,072$ emails between directed pairs of $N = 87,273$, senders and recipients of the former Enron corporation, disclosed as part of a fraud investigation[50].
Internet AS: This data set contains connections between autonomous systems (AS) collected by the CAIDA project (retrieved from http://www.caida.org). It corresponds to a time-stamped sequence of $E = 500,106$ directed connections between AS pairs, with a total of $N = 53,387$ recorded AS nodes. The time-stamps correspond to the first time the connection was seen.
APS citations: This dataset contains $E = 4,262,443$ time-stamped citations between $N = 425,760$ scientific articles published by the American Physical Society for a period of over 100 years (retrieved from http://journals.aps.org/datasets).

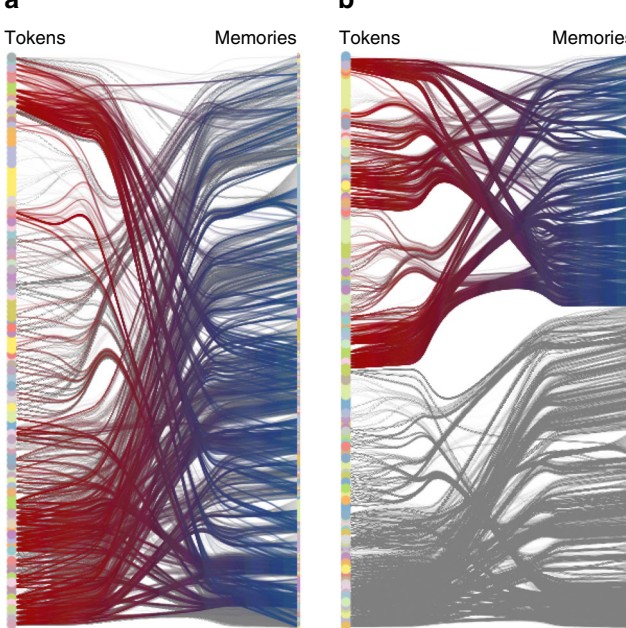

**Fig. 6** Markov model fit for a concatenated text. 'The texts are 'War and peace' by Leo Tolstoy and 'À la recherche du temps perdu', by Marcel Proust. Edge endpoints in *red* and *blue* correspond to token and memories, respectively, that involve letters exclusive to French. **a** Version of the model with $n = 3$ where no distinction is made between the same token in the different novels, yielding a description length $-\log_2 P(\{x_t\}, b) = 7,450,322$. **b** Version with $n = 3$ where each token is annotated with its novel, yielding a description length $-\log_2 P(\{x_t\}, b) = 7,146,465$

prosper.com loans: This data set corresponds to $E = 3,394,979$ time-stamped directed loan relationships between $N = 89,269$ users of the prosper.com website, which provides a peer-to-peer lending system (retrieved from http://konect.uni-koblenz.de/networks/prosper-loans).

Chess moves: This data set contains 38,365 online chess games collected over the month of July 2015 (retrieved from http://ficsgames.org/download.html). The games were converted into a bipartite temporal network where each piece and position correspond to different nodes, and a movement in the game corresponds to a time-stamped edge of the type piece → position. The resulting temporal network consists of $N = 76$ nodes and $E = 3,130,166$ edges.

Hospital contacts: This data set corresponds to a temporal proximity measurement of patients and health care workers in the geriatric unit of an university hospital[51]. A total of $N = 75$ individuals were voluntarily tracked for a period of 4 days, generating $E = 32,424$ proximity events between pairs of individuals.

Infectious sociopatterns: This data set corresponds to a temporal proximity measurement of attendants at a museum exhibition[52]. A total of $N = 10,972$ participants were voluntarily tracked for a period of 3 months, generating $E = 415,912$ proximity events between pairs of individuals.

Reality mining: This data set corresponds to a temporal proximity measurement of university students and faculty[53]. A total of $N = 96$ people were voluntarily tracked for a period of an entire academic year, generating $E = 1,086,404$ proximity events between pairs of individuals.

Beethoven's fifth symphony: A piano reduction of Beethoven's fifth symphony, extracted in MIDI format from the Mutopia project at http://www.mutopiaproject.org, represented as a sequence of $E = 4223$ notes of an alphabet of size $N = 63$.

**Code availability**. A free C++ implementation of the inference algorithm is available as part of the graph-tool Python library[45], available at http://graph-tool.skewed.de.

**Data availability**. The data sets analysed during the current study are available from their sources listed in the 'Data sets' section above, of from the corresponding author on reasonable request.

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

## Acknowledgements

We thank M. Buongermino Pereira and L. Bohlin for comments on the manuscript. M.R. was supported by the Swedish Research Council grant 2016-00796.

## Author contributions

Both authors conceived the project, performed the research, and wrote the paper. T.P.P. performed the analytical and numerical calculations.

## Additional information

Competing InterestsThe authors declare no competing financial interests.

