## [Peer Review File · Nature Communications]

Reviewers' comments:

Reviewer #1 (Remarks to the Author):

This paper proposes a parameterless Bayesian inference methodology analyzing trajectories on a network in order to deduce simultaneously a higher-order Markov model for those trajectories (on particular, deduce the order of the Markov chain) and communities for the nodes.

The work is extended to temporal networks, seen as a time series of edges, thus can be modelled by Markov chain.

The paper is well-written, although relatively difficult to read, and supported by interesting data sets.

Inferring a higher-order Markov model, including by Bayesian methodology, is not a new problem. The paper mentions Strelhoff et al. (PhysRev E 2007) in particular. On the minimum description length chosen here as a criterion, the methodology proposed here improves Strelhoff et al by 2 to 14% according to dataset.

Detecting communities taking into account the higher-order memory of trajectories has been studied by the second author in Nat Com 2014 [24].

The conceptual originality here resides in the simultaneous resolution of both steps, and the versatility to the details of the problem: discrete or continuous time, etc.

The methodology is similar to the one initiated by the first author for community and stochastic block model detection on networks (PhysRevLett 2013), hierarchies of block structures (PhysRevX 2014), multi-layer networks and time-varying networks (PhysRevE 2015).

It is adapted here to create a co-clustering of memories (list of current and past visited nodes) with the next node. The clustering of the next node is the desired community structure of the network.

The method has unintuitive aspects, as can be seen in the case $n=1$ (length of memory is one, thus a memory is simple a node).

In this case nodes are classified according two partitions as the partition on memories is also a partition on nodes. To rectify this difficulty, the authors propose to force those to partitions to be the same, specifically for $n=1$ (and only in this rectified case is equivalence with standard stochastic block model proved). Now we may ask, why not constraint the memories to be memories of node communities rather than being completely unrelated to the node communities? Eg for $n=2$ it is intuitive to consider that the next node depends on the current community and the past community, rather than on a arbitrary partition of edges (memories of 2 nodes). This would be an intuitive extension of stochastic block model to a multi-step framework. I understand that this flexibility certainly gets better compression results, by added flexibility. This complicatedness of the model does not matter if in the end it leads to an improvement of the quality of the communities.

However there is not much empirical justification of validity of the results. For instance no numerical comparison is made with the methodology developed by the second author in [24] (or: using Strelhoff et al to deduce the order of the model, on which [24] is used) . On the example of the US air flight itineraries, also studied in [24], the current methodology seems to reproduce exactly the same conclusions: memory matters as it allows to distinguish Atlanta from Las Vegas. Does it bring any new insight?

Method F brings the argument of disassortative communities that cannot be detected by Map formalism, it would be nice to see such structures in real data.

The application of temporal networks is counterintuitive. Take for instance emails in a large

company. The succession of emails ordered by their time stamps is unlikely to reveal any Markovian structure, because consecutive emails typically occur at very different places in the company, and are likely to be completely unrelated. So why look for such patterns? I can admit that the supplementary generative layer (partitioning nodes into C groups), currently introduced in an ad-hoc way, is in fact aimed at resolving that. A comment on that would be useful.

In conclusions, I think this paper combines existing ideas in a new way, resulting in a sophisticated methodology that may be (slightly or much) better than the existing methods to detect structures in dynamic networks. This would certainly be an excellent paper for a technical journal as it calls for more discussion and applications. In terms of maturity or novelty, in the present state it appears to me as borderline for a journal like Nature Communications, however.

Some minor comments:

Some sentences are exceedingly technical for the intended audience, not necessarily acquainted with the details of the Bayesian inference : for example 'multilevel Bayesian hierarchies, ie a sequence of priors and hyperpriors' and 'predictive likelihood of the validation set'.

'Token' is a surprising word to say 'node'. In many contexts, for example Petri nets, token are (non-random) walkers hopping from node to node, creating a confusion. Why not 'node', or 'state'?

When comparing with Strelhoff et al, it would be nice to be more specific about which method is used, as Strelhoff et al study several variants.

typo: 'markov chain' 'a generative processes'

'we always infer $BN=BM=C=1$ ': I assume the authors mean 'with high probability we infer $BN=BM=C=1$ ' since a random instance can generate any temporal network, thus any outcome can be found by the algorithm in principle.

Method E1, continuous time: since $\alpha = 0 = \beta$ is not properly normalized, we choose $\alpha = 1$ and $\beta = \sum \lambda / M$: is that an obviously natural choice? Why not α and β very small, for instance?

Figures are hard to read, due to small print. Especially Fig 4b. What is the color code in Fig 5 ?

Reviewer #2 (Remarks to the Author):

Summary of major claims

The authors introduce a novel community detection method which can be interpreted as a generalization of the stochastic block model for sequential data based on higher-order Markov chains. Extending a Bayesian approach to Markov model selection introduced by Strelhoff et al., they show how integrating group structures improves the modeling (and compression) of sequence data. For this, a parametrization of transition probabilities that takes into account (latent) community labels of both symbols and memory prefixes in an n-th order Markov chain is proposed. Finally, a non-parametric method to infer both the optimal Markov order and community memberships is developed.

The resulting method is applied to 14 data sets covering scenarios like text mining (character

sequences), temporal networks (sequences of time-stamped edges), flight and taxi trips (sequences of trip locations) as well as chess moves and music. The results show that the method consistently yields higher Markov orders and better compression performance compared to a commonly used Markov chain parametrization technique. It further provides insights into community structures in temporal networks which go beyond existing community detection techniques and it can be generalized to continuous-time and non-stationary data.

Novelty and interest to the community

The method proposed by the authors is interesting and the results are convincing. The work fits well into a recent stream of research on limitations of graph abstractions and community detection algorithms. It should thus be of broad interest not only to researchers in graph theory, statistical inference and network analysis, but also to interdisciplinary scholars interested in (biological) sequence analysis and time series modeling. Hence, it is likely to influence the thinking in different scientific fields. The combination of stochastic block model inference of community structures with higher-order Markov chain models of sequence data is both novel and innovative. I thus recommend it for publication in Nature Communications.

Detailed comments

While this work has the potential to advance our ability to detect patterns in sequence data, the following points should be addressed:

1) In the abstract and the discussion, the authors summarize the proposed method as a "variable-order hidden Markov chain model" (VOM). This does not seem to be correct, as the authors consider a model with fixed order n , while for a VOM the prefix length (i.e. the dimensionality of memory nodes) typically differs for different random variables. This does not seem to be the case here. If this interpretation of the method is wrong, the authors should better explain their methodology and highlight the relation to VOM techniques. If indeed multiple prefix lengths are used in the same model, the authors should include results showing which (or how many) of the nodes exhibit higher-order memory and which don't.

Unless the authors make a convincing case, the term "arbitrary-order hidden Markov chain" (or even more common: "higher-order Markov chain") used in the introduction seems to be more appropriate.

2) Section I of the manuscript is very technical and largely focuses on methodology rather than on results. Indeed, the choice of which parts of the methodology are moved to section III, and which are explained in section I seems rather arbitrary. Parts of the manuscript are unnecessarily hard to follow even for readers knowledgeable in statistical inference and more effort should be made to make at least section I accessibly also for non-experts and interdisciplinary researchers.

3) In section I A the authors correctly emphasize the risk of overfitting that comes with the use of higher-order Markov models. They argue that their method is able to infer the most appropriate Markov order without overfitting. Unfortunately, this essential aspect of the method is not explained well. The only reference is the comment that Eq. (11-12) serve as penalties that prevent overfitting. No further discussion about the choice of the penalty term and how the increase of model complexity for increasing n is accounted for in the description length is included. This should be explained in a better way. For this it may be helpful to move Eq. (9-12) to section III while including a less technical explanation in section I. The manuscript would further benefit from explicitly mentioning differences between the proposed method and the use of information criteria which are commonly applied for Markov order estimation.

4) The fact that the authors are able to infer Markov orders up to $n=4$ is noteworthy, especially considering that the underlying data sets are only moderately large (in the case of War and Peace

the sequence contains "only" about 3 Million tokens including artificially created "stop" tokens). Some remarks about the relation between (i) the length of the sequence, (ii) the number of communities of tokens and prefixes, and (iii) the detectable Markov order would be helpful. Specifically, it is striking that no higher-orders beyond $n=1$ have been detected for data sets where the number of tokens is comparably large compared to the sequence length (cf. Table II). This raises the question whether in these cases the absence of higher-order memory is actually due to a lack of statistics rather than an absence of correlations.

5) The motivation and the temporal prediction approach introduced in section I D, as well as the interpretation of Table III should be improved. Currently, this section feels disconnected to the rest of the manuscript and does not add much to support its main claims.

6) In the conclusion (and the abstract), the authors emphasize that their method requires no a priori knowledge about time scales in the data. The manuscript would benefit from a more thorough motivation of this problem, e.g. clearly defining what time scales the authors mean exactly. Arguably, in the modeling of sequences in section I C, the timing of events (and thus time scales that may or may not be relevant for the analysis) are neglected. While this is addressed in the continuous time extension in section I E, at least some prior knowledge about the waiting time distribution (e.g. Poisson vs. power law waiting times) is seemingly needed. Moreover, the method seems to rest on the assumption that waiting time distributions are uniform at least within groups, which may not hold in data where different individuals follow different activity patterns (despite being in the same group). A comment on this would be helpful.

7) As a final remark, the accessibility of the manuscript to non-experts in SBMs and statistical inference would largely benefit from a clarification of notation. For instance, n_r (supposedly referring to the number of nodes in group r) in Eq. (11) or b (supposedly the block assignment vector of tokens/memories) are not clearly defined and can only be inferred from the context. Replacing (rather simple) binomial coefficients, e.g. in (11) and (12), by a custom notation does not make it easier to follow the approach. In other parts, the notation is at least ambiguous like, e.g., the meaning of indices r and s in \mathcal{D}_r and \mathcal{D}_s in Eq. 8.

Minor remarks

- page 1, last paragraph: we extend this model to $_$ community-based temporal networks ...
- extra comma in Eq. (1)
- page 1: we want to infer the transition_ probabilities
- The notation in Eq. (6) is confusing. Consider defining e_s should be defined as $\sum_{r'} e_{r's}$
- the node labels in Fig. 2 are difficult to see, please increase them
- page 3, last paragraph: if we do not $_know_$ the order ...
- page 7, I D.: extra closing bracket in sentence
- page 9, caption Fig. 6: ... token $_and_$ memories ...
- page 9, caption Fig. 6: ... no distinction $_is_$ made ...

Point-by-point response to the referees' comments

Response to Referee 1

1. This paper proposes a parameterless Bayesian inference methodology analyzing trajectories on a network in order to deduce simultaneously a higher-order Markov model for those trajectories (on particular, deduce the order of the Markov chain) and communities for the nodes. The work is extended to temporal networks, seen as a time series of edges, thus can be modelled by Markov chain.

The paper is well-written, although relatively difficult to read, and supported by interesting data sets.

We thank the referee for the positive comments.

The method has unintuitive aspects, as can be seen in the case $n=1$ (length of memory is one, thus a memory is simple a node). In this case nodes are classified according two partitions as the

partition on memories is also a partition on nodes. To rectify this difficulty, the authors propose to force those to partitions to be the same, specifically for $n=1$ (and only in this rectified case is equivalence with standard stochastic block model proved). Now we may ask, why not constraint the memories to be memories of node communities rather than being completely unrelated to the node communities? Eg for $n=2$ it is intuitive to consider that the next node depends on the current community and the past community, rather than on a arbitrary partition of edges (memories of 2 nodes). This would be an intuitive extension of stochastic block model to a multi-step framework. I understand that this flexibility certainly gets better compression results, by added flexibility. This complicatedness of the model does not matter if in the end it leads to an improvement of the quality of the communities.

The referee interprets this aspect of the model somewhat different than we do. We view the co-clustering of memories and tokens not as a problem that needs to be rectified, but rather as an elegant solution that allows us to define a simple arbitrary-order Markov model.

To answer the specific question raised by the referee: If we would make the occurrence of the next token conditioned on the previous two communities, our transition probabilities would no longer be a matrix, but rather a third-order tensor. By increasing it to order n , we would have an $n + 1$ -order tensor. We believe such an approach would complicate the problem rather than simplify it. Moreover, the number of parameters would grow exponentially with the order of the markov chain.

Indeed, our co-clustering can be viewed as a low-dimensional and sparse higher-order tensor representation that relies on the clustering of the memories, and is sufficiently expressive. This can be seen by noting that a third-order tensor representing the transition probabilities $(r, s) \rightarrow t$ can be flattened as a matrix $u \rightarrow t$, where u represents the combination (r, s) . In our model we went further and divided the u combinations into classes (i.e. a sparse representation), thereby *reducing* the size of the transition matrix in accordance to the statistical evidence and structure in the data.

We have included the above point as a footnote in methods section of the manuscript.

However there is not much empirical justification of validity of the results. For instance no numerical comparison is made with the methodology developed by the second author in [24] (or: using Strelioff et al to deduce the order of the model, on which [24] is used).

The quantitative comparison between our work and Strelioff et al., which we detail in Table I, is possible because they are both inference procedures, and yield a description length or posterior likelihood that can be directly compared.

A direct quantitative comparison with Ref. [24] is not possible because: (i) That method

cannot determine the Markov order, which needs to be supplied as a parameter of the algorithm; *(ii)* it is not an inference procedure that yields a likelihood of the data; and *(iii)* it is based on a different representation of the data such that a one-to-one comparison of the resulting clustering is not feasible.

For example, using Strelhoff et al. to determine the Markov order and then using Ref. [24] to identify communities with long flow persistence times, as the referee suggests, would be an apple-and-pear comparison according to *(iii)* above. As we argue in “The many facets of community detection in complex networks” [Appl Netw Sci 2: 4. (2017)], and is also a main argument of “The ground truth about metadata and community detection in networks” [<https://arxiv.org/abs/1608.05878>], such a comparison across model classes can be misleading. We are simply asking different questions about the data.

In the revised manuscript, instead we have further developed the illustrative comparison between the methods when discussing the US Air flights data.

On the example of the US air flight itineraries, also studied in [24], the current methodology seems to reproduce exactly the same conclusions: memory matters as it allows to distinguish Atlanta from Las Vegas. Does it bring any new insight?

Again, rather than a full-fledged comparison with Ref. [24], our objective with this example is to illustrate similarities and differences between the two methods, and indeed what new insights our method provides.

First, we show that this dataset is better represented by a third-order Markov model, instead of a second-order model used in Ref. [24]. Second, the difference between Atlanta and Las Vegas is represented more clearly and differently by the hierarchical clustering of the memories: We see, for example, that the returns to either Atlanta or Las Vegas come from a set of memories that are divided into two overall groups each, before branching in smaller groups, which are divided according to each other airports they lead up to. In other words, the division between “transit” and “destinations” propagates all the way to the upper hierarchical levels of the memory partition. Third, our co-clustering also divides the airports into hierarchical categories. We can see, for example, that Atlanta is grouped with nearby Charlotte, NC at the first hierarchy level, and with Detroit, Minneapolis, Dallas and Chicago at the third level. This tells us that these serve as alternative destinations to itineraries that are similar to those that go thorough Atlanta. Likewise, Las Vegas is grouped together with alternative destinations Phoenix and Denver. None of the properties above are easily discernible with the approach of Ref. [24].

We have modified the section in question to make the above clearer.

Method F brings the argument of disassortative communities that cannot be detected by Map formalism, it would be nice to see such structures in real data.

The US Air Flights dataset show a good example of this, as we point out in the revised manuscript: The fact that the hierarchical division of memories and tokens is not symmetric shows that the inferred structures cannot be represented by an “assortative” model. This was also true for all other datasets that we analyzed.

The application of temporal networks is counterintuitive. Take for instance emails in a large company. The succession of emails ordered by their time stamps is unlikely to reveal any Markovian structure, because consecutive emails typically occur at very different places in the company, and are likely to be completely unrelated. So why look for such patterns? I can admit that the supplementary generative layer (partitioning nodes into C groups), currently introduced in an ad-hoc way, is in fact aimed at resolving that. A comment on that would be useful.

The reason why we look at such patterns is straightforward: We believe it is useful, even if it is not a perfect representation of the data. Note that the same argument can be used against using Markov chains in any context — not only networks. Nevertheless, they remain very useful models to understand many different types of data, such as DNA sequences and text, even though we know these are in fact not generated by Markov chains.

We should point out that we do, in fact, observe evidence for a Markovian structure in the email dynamics, contrary to the referee’s expectation, as we report in Table II. While this should not be interpreted as evidence of causal relationships (and we do not suggest this), it is an important — and statistically significant — dynamical pattern.

Although it is true that temporal edges (or tokens in general) occurring in sequence are not necessarily causally related, the context from which causality can be determined is often not available from data. For example, although we can certainly know that students in a high school are distributed in space, thus precluding far-away interactions from being causally related, this spatial information is simply not available in most datasets, such as the one we used.

However, we do in fact touch upon this issue in the manuscript, in section E.3 (Model extensions, nonstationarity). Although we focused on nonstationarity, mathematically the issue is the same: The Markov chain transitions depend on some hidden context (e.g. the spatial distribution of students, email threads). We showed in section E.3 that our approach can be extended in this direction, and that it essentially works in finding these hidden contexts. However, we left a more detailed exploration of this for future work.

We have modified section E.3 to make the above point.

In conclusions, I think this paper combines existing ideas in a new way, resulting in a sophisticated methodology that may be (slightly or much)

better than the existing methods to detect structures in dynamic networks. This would certainly be an excellent paper for a technical journal as it calls for more discussion and applications. In terms of maturity or novelty, in the present state it appears to me as borderline for a journal like Nature Communications, however.

With the revised abstract and introduction, together with the clarifications we have made throughout the manuscript thanks to the referees' comments, we hope that we have convincingly argued why and how our approach overcomes central limitations of current approaches, and that the scalable and publicly available code makes it highly useful in the many fields that deal with sequence and network data.

Some minor comments:

Some sentences are exceedingly technical for the intended audience, not necessarily acquainted with the details of the Bayesian inference : for example 'multilevel Bayesian hierarchies, ie a sequence of priors and hyperpriors' and 'predictive likelihood of the validation set'.

We have made a thorough revision of the manuscript, either avoiding overly technical terms, or explaining them where necessary.

'Token' is a surprising word to say 'node'. In many contexts, for example Petri nets, token are (non-random) walkers hopping from node to node, creating a confusion. Why not 'node', or 'state'?

We have to be careful, since in our paper there are many different "nodes": Memory nodes, "token" nodes, nodes of the network where a random walk is taking place, nodes of the temporal network, etc. "State" is also not right. The state of a n -order Markov chain is the last n tokens. Calling each of them "state" would be incorrect.

So, we have opted for the following terminology, designed to avoid ambiguity: Sequences are collections of tokens. The state of a n -order Markov chain is a sequence of n tokens. When dealing with temporal network, a token is an edge (pair of nodes).

The word "token" is not uncommon in the Markov chain literature (e.g. the Wikipedia article on hidden Markov chains uses it).

When comparing with Strelhoff et al, it would be nice to be more specific about which method is used, as Strelhoff et al study several variants.

The version we use here is the one given by Eq. 4 in our manuscript with noninformative priors. We state this clearly in the new manuscript.

typo: 'markov chain' 'a generative processes'

These have been corrected.

'we always infer $BN=BM=C=1$ ': I assume the authors mean 'with high probability we infer $BN=BM=C=1$ ' since a random instance can generate any temporal network, thus any outcome can be found by the algorithm in principle.

Our statement has been made more precise.

Method E1, continuous time: since $\alpha = 0 = \beta$ is not properly normalized, we choose $\alpha = 1$ and $\beta = \sum \lambda_{\bar{x}} / M$: is that an obviously natural choice? Why not α and β very small, for instance?

Making them very small would make the likelihood very small, since in the limit $\alpha \rightarrow 0$ and $\beta \rightarrow 0$ the normalization constant diverges.

The assumption $\alpha = 1$ and $\beta = \sum_{\bar{x}} \lambda_{\bar{x}} / M$ is a mild "empirical Bayes" assumption, where one imagines that the prior experience consists of a single data point ($\alpha = 1$) with a value given by the average in the data ($\beta = \sum_{\bar{x}} \lambda_{\bar{x}} / M$). This is not the only possible way to proceed, but the results will not depend strongly on this, because the data will eventually override any assumption made.

We have added a note on this in the text.

Figures are hard to read, due to small print. Especially Fig 4b. What is the color code in Fig 5 ?

The current version of the manuscript is for reviewing purposes only, and does not reflect the final layout. In case of publication, we will make sure the figure sizes will be increased.

The color code in Fig. 5 simply reflect the group labels.

Response to Referee 2

1. Summary of major claims

The authors introduce a novel community detection method which can be interpreted as a generalization of the stochastic block model for sequential data based on higher-order Markov chains. Extending a Bayesian approach to Markov model selection introduced by Strelisoff et al., they show how integrating group structures improves the modeling (and compression) of sequence data. For this, a parametrization of

transition probabilities that takes into account (latent) community labels of both symbols and memory prefixes in an n-th order Markov chain is proposed. Finally, a non-parametric method to infer both the optimal Markov order and community memberships is developed.

The resulting method is applied to 14 data sets covering scenarios like text mining (character sequences), temporal networks (sequences of time-stamped edges), flight and taxi trips (sequences of trip locations) as well as chess moves and music. The results show that the method consistently yields higher Markov orders and better compression performance compared to a commonly used Markov chain parametrization technique. It further provides insights into community structures in temporal networks which go beyond existing community detection techniques and it can be generalized to continuous-time and non-stationary data.

Novelty and interest to the community

The method proposed by the authors is interesting and the results are convincing. The work fits well into a recent stream of research on limitations of graph abstractions and community detection algorithms. It should thus be of broad interest not only to researchers in graph theory, statistical inference and network analysis, but also to interdisciplinary scholars interested in (biological) sequence analysis and time series modeling. Hence, it is likely to influence the thinking in different scientific fields. The combination of stochastic block model inference of community structures with higher-order Markov chain models of sequence data is both novel and innovative. I thus recommend it for publication in Nature Communications.

We thank the referee for the encouraging remarks, and the recommendation for publication.

Detailed comments

While this work has the potential to advance our ability to detect patterns in sequence data, the following points should be addressed:

- 1) In the abstract and the discussion, the authors summarize the proposed method as a "variable-order hidden Markov chain model" (VOM). This does not seem to be correct, as the authors consider a model with fixed order n , while for a VOM the prefix length (i.e. the dimensionality of memory nodes) typically differs for different random variables. This does not seem to be the case

here. If this interpretation of the method is wrong, the authors should better explain their methodology and highlight the relation to VOM techniques. If indeed multiple prefix lengths are used in the same model, the authors should include results showing which (or how many) of the nodes exhibit higher-order memory and which don't.

Unless the authors make a convincing case, the term "arbitrary-order hidden Markov chain" (or even more common: "higher-order Markov chain") used in the introduction seems to be more appropriate.

The point of our terminology was to emphasize that the order of the Markov chain is a parameter of the model that is recovered from data, instead of having to be determined a priori. But the referee is correct that it might be confused with models that have a varying order. We have replaced the term by "arbitrary-order" as suggested.

2) Section I of the manuscript is very technical and largely focuses on methodology rather than on results. Indeed, the choice of which parts of the methodology are moved to section III, and which are explained in section I seems rather arbitrary. Parts of the manuscript are unnecessarily hard to follow even for readers knowledgeable in statistical inference and more effort should be made to make at least section I accessibly also for non-experts and interdisciplinary researchers.

We have reorganized section I, moving most of the technical parts to section III, and explaining the approach at a qualitative level that should be more accessible to non-experts.

3) In section I A the authors correctly emphasize the risk of overfitting that comes with the use of higher-order Markov models. They argue that their method is able to infer the most appropriate Markov order without overfitting. Unfortunately, this essential aspect of the method is not explained well. The only reference is the comment that Eq. (11-12) serve as penalties that prevent overfitting. No further discussion about the choice of the penalty term and how the increase of model complexity for increasing n is accounted for in the description length is included. This should be explained in a better way. For this it may be helpful to move Eq. (9-12) to section III while including a less technical explanation in section I. The manuscript would further benefit from explicitly mentioning differences between the proposed method and the use of information criteria which are commonly applied for Markov order estimation.

We had omitted a detailed discussion on this because it is addressed at length in the cited references. However, we agree with the referee that it leaves a gap. We have now rewritten

this part such that this point can be clearly understood.

We followed the referee's suggestion and moved several equations to section III. We also included a comparison with other penalty methods.

4) The fact that the authors are able to infer Markov orders up to $n=4$ is noteworthy, especially considering that the underlying data sets are only moderately large (in the case of War and Peace the sequence contains "only" about 3 Million tokens including artificially created "stop" tokens). Some remarks about the relation between (i) the length of the sequence, (ii) the number of communities of tokens and prefixes, and (iii) the detectable Markov order would be helpful. Specifically, it is striking that no higher-orders beyond $n=1$ have been detected for data sets where the number of tokens is comparably large compared to the sequence length (cf. Table II). This raises the question whether in these cases the absence of higher-order memory is actually due to a lack of statistics rather than an absence of correlations.

The central difference between the datasets in Table I (simple sequences) from those in Table II (temporal networks) is in the number of tokens, more so than simply the length of the sequences — or rather in the relationship between these two quantities. In the case of temporal networks, the tokens are edges in the network. Hence, a temporal network with N nodes may have up to $O(N^2)$ tokens, representing each possible edge. An inspection of Table II reveals that the length of the sequence is far smaller than the total number of tokens (the alphabet), and hence most of them do not even occur in the sequence. This is completely different situation from Table I, where the length of the sequence is much larger than the size of the alphabet.

It is the above difference in the scaling scenario that is responsible for the fact that we encounter mostly $n = 1$ for these datasets; Any value $n > 1$ is not commensurate with the statistical evidence available, and represents an overfit. In view of this, we find that obtaining $n = 1$ to be quite remarkable, which is possible thanks to how we deal with the sparsity of the data by including the extra generative step where the nodes are divided into groups. Indeed, in many of these examples, if we remove this part of the model, we get a preferred model with $n = 0$, i.e. without temporal correlations whatsoever (this is not shown in Table I).

Note that the only exception to the above is the network of chess moves, where indeed the number of tokens $(76(76 - 1)/2 = 2850)$ is far smaller than the total length of the sequence ($\sim 3 \times 10^6$), and we find a better fit for $n = 2$.

We thank the referee for pointing out that this important point was not discussed at the appropriate level of detail. We have improved the discussion in the revised version of the

manuscript.

5) The motivation and the temporal prediction approach introduced in section I D, as well as the interpretation of Table III should be improved. Currently, this section feels disconnected to the rest of the manuscript and does not add much to support its main claims.

To clarify the motivation, we have now expanded this section. Our objective was two-fold: (i) To compare two different model selection criteria (unsupervised and supervised) and show that they generally agree with each other; and (ii) to emphasize that our approach not only provides a clustering of the data, but that it also provides a full-fledged model that is capable of generalizing from observations and provide predictions. The latter point is a major advantage when compared to alternative heuristic approaches.

6) In the conclusion (and the abstract), the authors emphasize that their method requires no a priori knowledge about time scales in the data. The manuscript would benefit from a more thorough motivation of this problem, e.g. clearly defining what time scales the authors mean exactly. Arguably, in the modeling of sequences in section I C, the timing of events (and thus time scales that may or may not be relevant for the analysis) are neglected. While this is addressed in the continuous time extension in section I E, at least some prior knowledge about the waiting time distribution (e.g. Poisson vs. power law waiting times) is seemingly needed. Moreover, the method seems to rest on the assumption that waiting time distributions are uniform at least within groups, which may not hold in data where different individuals follow different activity patterns (despite being in the same group). A comment on this would be helpful.

What we mean is that we can avoid what a large fraction of other approaches impose: (i) A separation of the sequence into intervals (temporal bins or layers), where inside each interval the temporal evolution is ignored (eg. Refs [11-19]); and (ii) an a priori Markov order (Ref. [24]).

Other than assuming a n -order Markov chain (which, we admit, is in itself an assumption about the temporal evolution of correlations), we make no other explicit assumption about time scales. Although it is true that the continuous-time model extension requires a prior on the waiting times and assumes that these are uniform inside the groups — and hence the referee is correct that this involves *some* assumption of time-scales — at the same time we are dealing with a mixture model, which will combine these uniform distribution in nonuniform ways. So, for example, if a group of memories in the data have different waiting times (as the referee considered), what will happen is that the method will use this information to split this group in two or more groups, according to the waiting times. The key point here is that this does not need to be determined a priori, since our inference procedure will find it.

We have modified the text to make the above points clear.

7) As a final remark, the accessibility of the manuscript to non-experts in SBMs and statistical inference would largely benefit from a clarification of notation. For instance, n_r (supposedly referring to the number of nodes in group r) in Eq. (11) or b (supposedly the block assignment vector of tokens/memories are not clearly defined and can only be inferred from the context. Replacing (rather simple) binomial coefficients, e.g. in (11) and (12), by a custom notation does not make it easier to follow the approach. In other parts, the notation is at least ambiguous like, e.g., the meaning of indices r and s in \mathcal{D}_r and \mathcal{D}_s in Eq. 8.

We have carefully revised the text, and made sure the notation is explicitly defined.

We note, however, that the notation $\binom{S}{r}$ for the multiset coefficient is not custom, and is in fact standard in the literature and textbooks (see e.g. Stanley (1997) “Enumerative Combinatorics”). The reason we use it in equations such as 11 and 12 is precisely to make them easier to read, since they avoid repetition of terms, and also they reveal the combinatorial reasoning behind the expressions. We have made a small note on this in the text.

Minor remarks

- page 1, last paragraph: we extend this model to `_ community-based temporal networks ...`
- extra comma in Eq. (1)
- page 1: we want to infer the transition `_ probabilities`
- The notation in Eq. (6) is confusing. Consider defining e_s should be defined as $\sum_{r'} e_{r's}$
- the node labels in Fig. 2 are difficult to see, please increase them
- page 3, last paragraph: if we do not `_ know_ the order ...`
- page 7, I D.: extra closing bracket in sentence
- page 9, caption Fig. 6: `... token _and_ memories ...`
- page 9, caption Fig. 6: `... no distinction _is_ made ...`

We have corrected these typos in the new version.

REVIEWERS' COMMENTS:

Reviewer #1 (Remarks to the Author):

I believe the paper, as interesting as ever, is now more readable thanks to the substantial efforts of the authors.

Regarding responses to my previous comments: I agree that the 'tensor' approach is just an alternative, probably more costly in algorithmic terms for $n > 1$, although described by the authors as 'slightly better' for 'many datasets' in the case $n = 1$. My point is that these are both potentially useful models. I also agree when they authors say we should not compare apple and pears (meaning, this paper and others like [24]), and this model is merely useful to describe the email dataset, even though it does not capture the full structure.

This pragmatic viewpoint could be more present in some parts of the manuscript, most importantly the abstract, introduction and conclusions. For example "The model infers not only the optimal Markov order but also the number of groups because the underlying arbitrary-order Markov chain model is nonparametric." (Conclusions, and see similar sentence in Abstract and Introduction) suggests that because the method is nonparametric, it provides the best Markov order and (best) number of groups. But, taking older methods of community detection, modularity maximization and map equation are both nonparametric, principled, data-driven methods that would often recommend different 'best' numbers of groups. Both would be correct with respect to the application in different circumstances. The neutral statement here is that the Markov order and number of groups are not an input but an output of the method. Of course the output is only correct or optimal with respect to a specific objective function, appropriate in certain circumstances and not in others (as the authors write in their letter and in Methods C and G). The minimum description length framework is indeed elegant and efficient, but by no means without ambiguity (as for instance Method C, E1 or the discussion on hierarchical grouping show) and by no means the only possible nonparametric framework. I would strongly recommend a softer formulation in the key parts of the manuscript.

Similarly: 'In particular, if there is no structure in the data—either through fully random dynamics or a lack of large-scale structure in the network—our method will [...] conclude that the data lack structure.' To conclude that the 'there is no structure in the real-life data' one needs a definition of 'structure', and there are many choices for that (e.g. through one or another statistical test). If the authors mean their own framework to define structure, then the sentence is tautological. But perhaps by lack of large structure, they mean that the data is not real but generated by an Erdos Renyi process, in which case their method will find a single group with high probability, and something similar for the dynamics, as they suggest later with shuffling sequences. Of course that is a more restrictive but more widely acceptable statement.

Other minor comments:

Fig3: it took me a while to discover the 'overlaid hierarchical' groupings in the picture, this should be optimized in a later version of the picture.

For that example, it seems that the choice of hierarchical groupings is made, that is only briefly explained in the text. It should be made explicit for each example whether the single grouping or hierarchical grouping is used (in which case: what is the 'number of groups' that is given as an output of the method? the bottom layer?)

Fig 5: as far as I understand the colors that code the groups have no correspondance in (a) and

(b) (despite the same color range being used), and no correspondance with another picture, and do not materialize a continuous quantity (as in a heat map), therefore seem useless and even confusing.

typos: the the, a remarkable that, as well as shape the networks themselves (check syntax of whole sentence)

Reviewer #2 (Remarks to the Author):

I comment on the revised version of this manuscript, specifically focusing on changes made by the authors in response to my earlier remarks.

The manuscript has been revised substantially and I believe that the revisions have considerably improved both the clarity and the accessibility of the work. I concur with the authors' comments in the rebuttal letter regarding the use of terminology, and I agree with the clarifications that have been made. The new structure and motivation of the article (especially the more concise and less technical description in section I.B) further help readability and allow to better judge the contribution of this work. The authors have also clarified the use of arbitrary- vs. variable-order Markov chain terminology, which will should help readers to better understand the approach.

I am particularly satisfied with the clear separation between what are the key findings and what are the underlying methods (now moved to section III) identified by the authors. I think that this has massively improved the manuscript. I have finally verified that the suggested clarifications of mathematical notation (e.g. missing definitions of some variables) mentioned in my earlier review have been addressed.

In summary, I stand by my judgment that this works makes an inportant contribution to the data-driven study of networked systems. Highlighting limitations of current network-analytic methods, it provides a practical new approach to study community structures in time series data. This work should be of interest for a wide range of researchers and I thus recommend it for publication in Nature Communications in its current form.

Point-by-point response to the referees' comments

Response to Referee 1

1. I believe the paper, as interesting as ever, is now more readable thanks to the substantial efforts of the authors.

Regarding responses to my previous comments: I agree that the 'tensor' approach is just an alternative, probably more costly in algorithmic terms for $n > 1$, although described by the authors as 'slightly better' for 'many datasets' in the case $n = 1$. My point is that these are both potentially useful models. I also agree when they authors say we should not compare apple and pears (meaning, this paper and others like [24]), and this model is merely useful to describe the email dataset, even though it does not capture the full structure.

We thank the referee for the positive comments about the revised manuscript.

This pragmatic viewpoint could be more present in some parts of the manuscript, most importantly the abstract, introduction and conclusions. For example "The model infers not only the optimal Markov order but also the number of groups because the underlying arbitrary-order Markov chain model is nonparametric." (Conclusions, and see similar sentence in Abstract and Introduction) suggests that because the method is nonparametric, it provides the best Markov order and (best) number of groups. But, taking older methods of community detection, modularity maximization and map equation are both nonparametric, principled, data-driven methods that would often recommend different 'best' numbers of groups. Both would be correct with respect to the application in different circumstances. The neutral statement here is that the Markov order and number of groups are not an

input but an output of the method. Of course the output is only correct or optimal with respect to a specific objective function, appropriate in certain circumstances and not in others (as the authors write in their letter and in Methods C and G). The minimum description length framework is indeed elegant and efficient, but by no means without ambiguity (as for instance Method C, E1 or the discussion on hierarchical grouping show) and by no means the only possible nonparametric framework. I would strongly recommend a softer formulation in the key parts of the manuscript.

Similarly: 'In particular, if there is no structure in the data-either through fully random dynamics or a lack of large-scale structure in the network-our method will [...] conclude that the data lack structure.' To conclude that the 'there is no structure in the real-life data' one needs a definition of 'structure', and there are many choices for that (e.g. through one or another statistical test). If the authors mean their own framework to define structure, then the sentence is tautological. But perhaps by lack of large structure, they mean that the data is not real but generated by an Erdos Renyi process, in which case their method will find a single group with high probability, and something similar for the dynamics, as they suggest later with shuffling sequences. Of course that is a more restrictive but more widely acceptable statement.

To avoid any ambiguity, we have made our claims more specific. For example, the last sentence of the abstract now reads

“We base our method on an arbitrary-order Markov chain model with community structure, and develop a nonparametric Bayesian inference framework that identifies the simplest such model that can explain temporal interaction data.”

and the corresponding sentence in the discussion reads

“The model does not require the optimal Markov order or number of groups as inputs, but infers them from data because the underlying arbitrary-order Markov chain model is nonparametric.”

About structure in the introduction, we now write:

“In both cases, we employ a nonparametric Bayesian inference framework that allows us to select, according to the statistical evidence available, the most parsimonious model among all its variations. Hence we can, for example, identify the most appropriate Markov order and the number of communities without overfitting. In particular, if the dynamics on or of a network are random, our method will not identify any spurious patterns from noise but conclude that the data

lack structure. As we also show, the model can be used to predict future network dynamics and evolution from past observations. Moreover, we provide publicly available and scalable code with log-linear complexity in the number of nodes independent of the number of groups.”

Fig3: it took me a while to discover the 'overlaid hierarchical' groupings in the picture, this should be optimized in a later version of the picture.

We have enhanced the contrast of the figure.

For that example, it seems that the choice of hierarchical groupings is made, that is only briefly explained in the text. It should be made explicit for each example whether the single grouping or hierarchical grouping is used (in which case: what is the 'number of groups' that is given as an output of the method? the bottom layer?)

We use the hierarchical model for all examples, without exceptions. The number of groups always refers to the bottom layer.

We have made this more explicit in the current manuscript version.

Fig 5: as far as I understand the colors that code the groups have no correspondance in (a) and (b) (despite the same color range being used), and no correspondance with another picture, and do not materialize a continuous quantity (as in a heat map), therefore seem useless and even confusing.

The colors match the numeric values of the memory groups.

typos: the the, a remarkable that, as well as shape the networks themselves (check syntax of whole sentence)

Thanks, we have corrected the typos.